# Characterizing the evolution and phenotypic impact of ampliconic Y chromosome regions

Elise A. Lucotte [1,2] ✉, Valdís Björt Guðmundsdóttir[3,4], Jacob M. Jensen[1], Laurits Skov[1], Moisès Coll Macià [1], Kristian Almstrup [5], Mikkel H. Schierup [1], Agnar Helgason [3,4] ✉ & Kari Stefansson [3,6]

A major part of the human Y chromosome consists of palindromes with multiple copies of genes primarily expressed in testis, many of which have been claimed to affect male fertility. Here we examine copy number variation in these palindromes based on whole genome sequence data from 11,527 Icelandic men. Using a subset of 7947 men grouped into 1449 patrilineal genealogies, we infer 57 large scale de novo copy number mutations affecting palindrome 1. This corresponds to a mutation rate of $2.34 \times 10^{-3}$ mutations per meiosis, which is 4.1 times larger than our phylogenetic estimate of the mutation rate ($5.72 \times 10^{-4}$), suggesting that de novo mutations on the Y are lost faster than expected under neutral evolution. Although simulations indicate a selection coefficient of 1.8% against non-reference copy number carriers, we do not observe differences in fertility among sequenced men associated with their copy number genotype, but we lack statistical power to detect differences resulting from weak negative selection. We also perform association testing of a diverse set of 341 traits to palindromic copy number without any significant associations. We conclude that large-scale palindrome copy number variation on the Y chromosome has little impact on human phenotype diversity.

The Y chromosome has undergone extraordinarily rapid evolution in mammals[1,2]. One contributing factor is the absence of genetic recombination (except in the small pseudoautosomal regions), which is an important mechanism for purging deleterious mutations in autosomal chromosomes. Despite the extensive degeneration of the Y chromosome and its apparent tolerance for large-scale rearrangements, it does carry some functional genes with important functions, which are essential for male fertility[3]. Of particular interest are the ampliconic genes, which are expressed in testis and organised in large inverted repeats called palindromes with very high (99.97%) sequence identity between palindromic arms[4]. The male-specific region of the Y chromosome (MSY) contains eight key palindromes, with a well-established numerical nomenclature: P1 to P8[5] and most of the ampliconic genes are contained within a large region called Azoospermia factor c (*AZFc*, Fig. 1A). *AZFc* is composed of P1, P2 and a part of P3 (Fig. 1), which in turn can be divided into the smaller repeated sequences called amplicons[6].

The inverted-repeat structure of the palindromes promotes pairing between their arms, which provides a mechanism for the rescue of deleterious mutations in the ampliconic genes through gene conversion[4,7,8]. Ectopic pairing and recombination between direct repeats within ampliconic regions can lead to structural mutations, such as deletions and duplications, which are recurrent in human populations[1,7,9,10], especially in the *AZFc* region.

[1]Bioinformatics Research Centre, Aarhus University, Dk-8000 Aarhus C., Denmark. [2]Ecologie Systematique et Evolution, CNRS, Université Paris-Saclay, AgroParisTech, 91198 Gif-sur-Yvette, France. [3]deCODE genetics/Amgen Inc., 101 Reykjavik, Iceland. [4]Department of Anthropology, University of Iceland, 101 Reykjavik, Iceland. [5]Department of Growth and Reproduction, Rigshospitalet, Copenhagen, Denmark. [6]Faculty of Medicine, School of Health Sciences, University of Iceland, 101 Reykjavik, Iceland. ✉e-mail: elucotte@gmail.com; agnar.helgason@decode.is

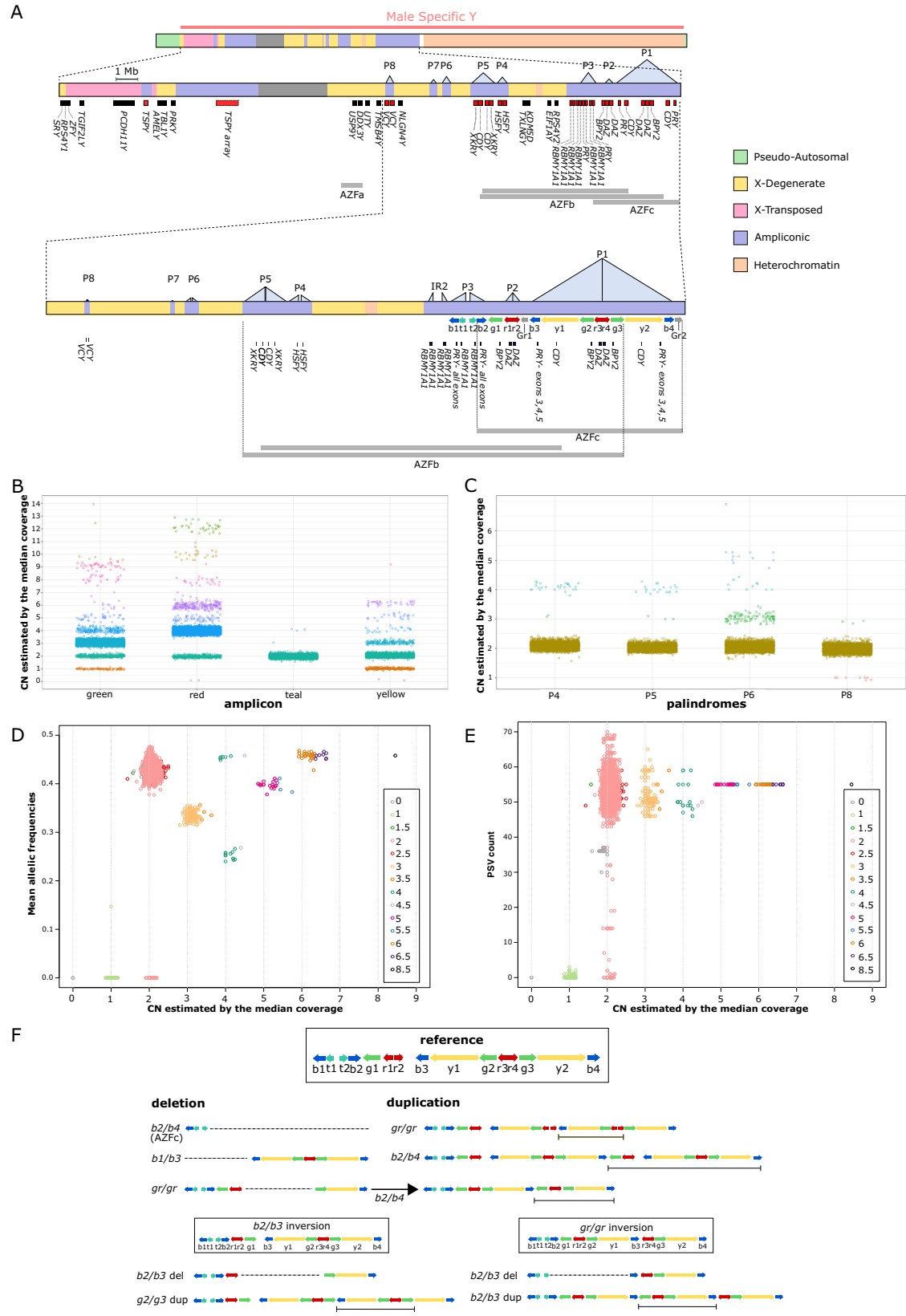

Previous studies have provided some evidence for selective constraints on ampliconic copy number (CN) variation[1,11]. One indication of this is that most humans carry the reference CN, with deviations typically due to recent mutations recurrently arising across the different haplogroups[1,9,12]. If purifying selection acts to constrain the variation of CN variants, then phylogenetic estimates of the mutation rate, the phylogenetic mutation rate, are expected to be lower than pedigree-based estimates. In this study, we use whole genome sequence (WGS) data from 11,525 Icelandic men, with an average of 15X coverage for the Y chromosome, to provide both phylogenetic (N = 891) and pedigree-based estimates (N = 7947 in 1449 patrilines) of the mutation rate of ampliconic CN genotypes in humans. Differences

**Fig. 1 | Copy number distribution of the Y chromosome palindromes in the Icelandic population. A** Y chromosome ampliconic regions. Top is a rectangular schematic of the entire Y chromosome. Below is a zoomed version of the mappable part of the male specific region of the Y chromosome, with the palindromes represented by labelled light blue triangles and the positions of genes shown below. Genes in red are known to be ampliconic and the grey shaded areas represent the centromere. Third is a further zoomed version of the ampliconic regions, again with the positions of palindromes and genes shown, and now also with amplicons shown as labelled coloured arrows. In the reference sequence of the human Y chromosome (NCBI build 38), the yellow amplicon (573,272 bp) has two copies (unique to palindrome P1), the green amplicon (307,076 bp) has three copies (two in P1 and one next to P2), the red amplicon (127,091 bp) has four copies (two in P1 and two in P2), the blue amplicon (167,703 bp) has four copies (two in P1 and two in P3), and the teal amplicon (115,689 bp) has two copies (unique to P3). **B** Copy number of the amplicons located in palindromes 1, 2 and 3. **C** Copy number of palindromes 4, 5, 6 and 8. **D** Mean minor allele read frequency (MARF) of paralogous sequence variants per male across the yellow amplicon shown in relation to copy number. **E** Number of paralogous sequence variants passing the filters used to calculate the MARF values presented in (**D**), shown in relation to copy number. **F** The structure of amplicons in the reference sequence and some of the known structural variants identified by previous studies, characterised deletions and amplification within the *AZFc* region. For deletions, the missing region is indicated with dotted lines. For duplications, the added region is indicated with a line delimiting the interval of the duplication.

between the phylogenetic mutation rate and the patriline mutation rates can then inform us about the extent of purifying selection acting on the ampliconic CN variants.

A change of CN within palindromes can lead to a change of CN of ampliconic genes, which could have a phenotypic impact. Large deletions of parts of the Y chromosome are thought to cause infertility (*AZFa*, *b* and *c*), while partial deletions of the *AZFc* region have been associated with a variable testicular phenotype, associated with low sperm counts and male infertility[6,13–15]. Testing of Y-microdeletions is routinely performed in the clinical work-up of men presenting with infertility because Y-microdeletions are the second most frequent genetic cause known to be associated with male infertility[16]. However, the estimated effect of partial deletions of *AZFc* on sperm quality is not consistent across studies, haplogroups or populations (reviewed in[17–19]). While the *b2/b3* and *gr/gr* deletions (both partial *AZFc* deletions) have been associated with lower fertility in some studies, the former is fixed in haplogroup N and the latter is fixed in haplogroup D2 and Q1, indicating either negligible impact on fertility or an effect that was largely countered by some other lineage-specific compensatory factors.

Genome-wide association studies (GWAS) have vastly increased our knowledge of the impact of genotypes on human phenotypes during the last decade[20]. However, these studies have largely ignored the Y chromosome. Apart from male fertility, there is a striking lack of phenotypic associations with sequence and structural variants from the Y chromosome[21]. In an attempt to shed further light on this issue, we explored the impact of CN variation in palindromes on a set of 341 phenotypes in the Icelandic population.

In this study, we detect 57 large scale de novo copy number mutations affecting palindrome 1 in patrilines, yielding a remarkably high estimate of the de novo mutation rate ($2.34 \times 10^{-3}$ mutations per meiosis). In contrast, our phylogenetic estimate of the mutation rate ($5.72 \times 10^{-4}$) is 4.1 times smaller than our patriline estimate, suggesting that de novo mutations on the Y are lost faster than expected under neutral evolution. Yet, we do not find differences in fertility among sequenced men associated with their copy number nor do we find any significant association between a diverse set of 341 traits and palindromic copy number. However, we show that we lack statistical power to detect differences resulting from weak negative selection. Finally, we do find indirect evidence of weak negative selection (1.8% against non-reference carriers) using simulations based on the discrepancy between the de novo mutation rate and the observed frequency of copy number variations in the Icelandic population.

## Results

### Population variation
We first estimated the CN genotype for each Y chromosome palindrome in 11,527 Icelandic males. We used a sequence coverage-based approach to call ampliconic CN genotypes in males, as recommended in the literature[22,23]. Sequence reads were mapped to a custom reference genome containing the proximal arms of the eight palindromes (with ampliconic genes masked) and the single copy genes of the Y chromosome (Supplementary Tables 1–3, Supplementary Fig. 1, Supplementary Methods section I). To estimate the CN of the target regions in each male, we calculated the median coverage across positions and standardised by the median of the median coverage of 17 single copy genes from the Y chromosome, which amount to 2 Mb of sequence[7,9].

The amplicons are grouped into six color-coded sub-unit families (yellow, green, red, blue, teal, and grey) based on sequence similarity[6]. The human Y chromosome reference sequence has an array of 17 ampliconic units: 2 yellow, 3 green, 4 red, 4 blue, 2 teal and 2 grey (their order is shown in Fig. 1A). The grey amplicon was omitted from our analysis. The blue amplicon one (b1 167,703 bp) and P7 (23,149 bp) were omitted because their CN genotypes could not be called accurately. Especially for the blue amplicon our estimates show a large variance and poorly defined CN clusters compared to the other amplicons (Supplementary Figs. 2 and 3, Supplementary Tables 1–3).

When unambiguous CN genotypes for a target region could not be determined (Supplementary Methods section I.5 and 6), we assigned a range of the two adjacent copy numbers to an individual (for example, '2-3' when the CN was estimated to be midway between 2 and 3). For the yellow amplicon, 35 males (0.3% of the whole sample) were assigned an ambiguous CN, while 169 (1.5%) and 364 (3.2%) males were assigned ambiguous CN for the green and red amplicons, respectively (see Supplementary Table 4 for the other regions). To assess the reliability of our CN calls, we examined 91 males who were sequenced at least twice from different tissue samples (buccal and/or blood samples). No inconsistencies were detected for the yellow and teal amplicons and for P4, P5, P6 and P8 (Binomial 95% CI= [0.0%, 5.9%], see Supplementary Methods section I.7). For the green and red amplicons, we detected 1 and 2 inconsistencies (one copy differences), respectively, yielding discrepancy rates of 1.3% (Binomial 95% CI = [0.1%, 8.1%]) and 2.6% (Binomial 95% CI = [0.4%, 9.9%]), respectively.

In line with previous studies[1,7,9,10], the CN variation in the Icelandic males is mostly located in the *AZFc* region covering P1, P2 and part of P3 (Fig. 1B, C, Supplementary Fig. 2, Supplementary Tables 5 and 6). We find that 8.6% of individuals (992 males) harbour non-reference CN variants in the *AZFc* region, most of them encompassing both P1 and P2 (887 males, Supplementary Table 6). Only 1.3% of males (151 males) carry non-reference CN variants in P4, P5, P6 or P8 (Supplementary Fig. 3, Supplementary Table 7). Most of these are in P6 (1%, 117 males) or P4 together with P5 (21 males, 0.2%). P4 + P5 events mostly co-occur with CN variation in P1 (20 individuals, Supplementary Table 8). We observe non-reference CN variants in each of the most common haplogroups (Supplementary Figs. 2 and 3), indicating that they arose independently due to multiple recurrent CN mutations.

We performed two different validation analyses based on a subset of 9162 individuals, whose sequence reads were mapped independently to the entire reference genome (build 38)[24]. For these analyses, reads mapping to paralogous positions were combined (see Supplementary Methods section I.9). First, we corroborated the CN genotype calls for the yellow amplicon using read frequencies at paralogous

positions with allelic variation – hereafter referred to as paralogous sequence variants (PSVs) (Supplementary Data 1, Supplementary Methods section I.9). The minor allele read frequency (MARF), i.e. the proportion of reads carrying the minor allele, was calculated for each PSV. The MARF at PSVs has an expectation of min(n0,n1) / N, where N is the number of yellow amplicon copies carried by a male and n0 and n1 are the number of copies with the ancestral (0) and derived (1) alleles, respectively, in a pseudo-heterozygous genotype. We found a clear relationship between MARF and CN, with clusters of individuals around expected values (Fig. 1D, E). For example, all individuals with two or six copies, have MARF values around 0.5 (half of the copies carry the minor allele), while individuals with three copies have MARF values around 0.33 (one of three copies carries the minor allele). Interestingly, the 19 males with four copies of the yellow amplicon can be divided into two subgroups based on MARF, with 12 showing a value of 0.25 (one of four copies carries the minor allele) and 7 with 0.5 (two of four copies carries the minor allele). It is likely that the two MARF clusters reflect differences in the mutational events that led to four copies of the yellow amplicon. We did not observe more than one MARF value for any other copy number. Thus, all 33 males with six copies have MARF = 0.5 (possible values are 0.167, 0.33 and 0.5) and all 22 males with five copies have MARF = 0.4 (possible values are 0.2 and 0.4).

In the second validation analysis based on the independent mapping of reads, we used relative sequence depth (rdepth) as a proxy for CN, this time using the average sequence depth per male across the single copy X-degenerate regions[5] (Fig. 1A) for normalisation, a total of 8,974,361 positions (Supplementary Methods section I.9.b). A comparison of these CN estimates with those estimated as part of the main analysis revealed a mismatch rate of 0.0003 for the median and 0.001 for the mean, excluding individuals categorised as 0 CN owing to their carrying a partial deletion of the yellow amplicon (see Supplementary Methods section I.6.a, Supplementary Fig. 4). A mismatch was recorded when our independent estimate of rdepth differed by more than 0.5 copies from our original estimate (see Supplementary Table 9). Overall, these two validation analyses provide good support for the amplicon CN genotypes used in this study.

Finally, we provide an additional validation of our CN calling by applying our method to 13 individuals of the 1000 Genomes project, who were examined in the Teitz et al. study[1] (Supplementary Table 10). The CN estimates show complete consistency between both approaches.

We classified amplicon CN genotypes using a nomenclature devised to describe recurrent deletions and duplications in the *AZFc* region of the Y chromosome (reviewed in[17], Fig. 1F, Supplementary Methods section I.1). To classify the CN variations detected in the population, we performed a density-based clustering (see Supplementary Methods section I.10 and I.11) that provided a CN estimate based on the median of the medians over 1 kb windows along the amplicon. Out of the 1011 Icelandic men with non-reference CN variants, 783 (77.4%) could be assigned to previously described recurrent CN events with breakpoints in specific amplicons (Supplementary Figs. 5 and 6, Table 1). Of the remaining 228 individuals, 212 could be classified into 4 clusters of previously unreported combinations of deletions and duplications (Supplementary Figs. 5, 7, and Supplementary Methods section I.11).

## Mutation rate from patrilines

We estimated the rate of de novo CN mutations for amplicons in the *AZFc* region (P1, P2 and P3) using a subset of 7947 Icelandic males, who could be grouped into 1,449 autosomal genotype-verified patrilineal genealogies with 24,380 meiotic transmissions (Table 2, Supplementary Tables 11 and 12, Supplementary Methods section II.1). We defined de novo CN mutations as differences in amplicon genotypes within patrilines (Supplementary Methods section II.2) and used parsimony

to assign mutations to specific patriline branches (Fig. 2A, B). Table 2 shows the number of de novo mutations identified for each amplicon. The per-generation mutation rate ($\mu_g$) for the yellow amplicon is estimated to be $2.34 \times 10^{-3}$ (95% Binomial CI= [$1.77 \times 10^{-3}$, $3.03 \times 10^{-3}$], 1 per 428 meioses), based on the detection of 57 events in 24,380 meiosis.

Previous studies have shown that mutations typically involve multiple amplicons[1,7,9]. In accordance with this, all but one event detected in the yellow amplicon also involve the green and red amplicons, indicating that most de novo events span both P1 and P2 (Supplementary Table 13). Very few de novo events are seen to affect only the green ($N = 3$) and red ($N = 1$) amplicons, and they are uncertain due to the higher error rate of CN calls for the smaller green and red amplicons (see below). The CN mutation rates in other palindromes were much lower (Table 2), with only two events detected for the teal amplicon (P3, $\mu_g = 8.20 \times 10^{-5}$, 95% CI= [$9.93 \times 10^{-6}$, $2.96 \times 10^{-4}$]), one event each in P4 and P6 ($\mu_g = 4.10 \times 10^{-5}$, 95% CI = [$1.04 \times 10^{-6}$, $2.29 \times 10^{-4}$]) and two events in P5 ($\mu_g = 8.20 \times 10^{-5}$, 95% CI = [$9.93 \times 10^{-6}$, $2.96 \times 10^{-4}$]). We identified one very large duplication event of around 2.35 Mb that encompasses P1, P2, P3, P4 and one arm of P5.

To validate our estimates of $\mu_g$, we grouped de novo mutations into two classes: 1) single descendant (SD) events (observed in only one sequenced male in a patriline) and 2) multiple descendant (MD) events (observed in at least two sequenced males in a patriline). We note that MD events are less likely to be false positives, because of the intrinsic replication of observing the same mutated allelic state independently in more than one patriline member. To estimate the mutation rates for these two classes, we counted the total number of meiotic transmissions across all patrilines that could lead to SD (15,799 meioses) and MD (8581 meioses) events, respectively. For the yellow amplicon, $\mu_g = 1.98 \times 10^{-3}$ for MD events (95% CI= [$1.15 \times 10^{-3}$, $3.17 \times 10^{-3}$]) and $2.53 \times 10^{-3}$ for SD events (95% CI = [$1.81 \times 10^{-3}$, $3.45 \times 10^{-3}$]), which are not statistically different (Fisher exact *p*-value = 0.49). Interestingly, there were slightly more deletions in SD (60%, 24 of 40) than MD (23.5%, 4 of 17) events (Fisher exact test *p*-value = 0.027) (Table 2). Although false-positive de novo CN mutations are more likely to appear as SD events, there is no reason to expect a bias towards deletions. As detrimental mutations are less likely to be transmitted to multiple individuals and across multiple generations in patrilines, this could be taken as weak evidence for negative selection acting against deletions.

To further validate the de novo call accuracy, we assessed the false positive rate of the yellow amplicon using CN genotypes from five additional men with WGS data, who were not included in the original analyses and who are descendants of lineages with inferred de novo events. All four de novo events (2 MD and 2 SD) that could be assessed with these men were confirmed (see Supplementary Methods section I.7). Moreover, we compared the distribution of the median coverage in 1 kb windows for each man in the patrilines where de novo events were detected. In all cases, the distribution of coverage was consistent with the inferred de novo event (see an example in Fig. 2C and Supplementary Fig. 8), such that males with a de novo event had distinct patterns of coverage when compared with other males from the same patriline. In addition, we note that the median CN of individuals with de novo events fall within the range of other individuals with the same estimated CN that were not due to de novo events (Fig. 2D).

We found that most of the events detected in patrilines involve the deletion or duplication of whole palindromes (59 of 62 events, 95.2%), suggesting that they are caused by erroneous resolution of pairing between paralogous amplicons, perhaps during attempted gene conversions. We used a hidden Markov model (HMM), where the hidden states are the CN, estimated for 5 kb (large palindromes) or 1 kb (small palindromes) windows (Supplementary section I.5.b). Only two partial events were detected in the yellow amplicon (1 SD and 1 MD event), both duplications. In addition, one of the de novo events that

**Table 1 | The amplicon CN profiles of previously reported deletions and amplifications, with information about their frequency in Icelanders (our data) and Poles (Rozen et al. 2012)**

| Event | Iceland | | Poland * | | Copy number | | | | |
|---|---|---|---|---|---|---|---|---|---|
| Type | Count | Percentage | Count | Percentage | yellow | green | red | teal | blue |
| reference | 10516 | 91.23% | 4445 | 95.16% | 2 | 3 | 4 | 2 | 4 |
| b2/b4 del | 2 | 0.02% | 2 | 0.04% | 0 | 0 | 0 | 2 | 1 |
| b1/b3 del | 1 | 0.01% | 2 | 0.04% | 2 | 2 | 2 | 0 | 2 |
| gr/gr del | 502 | 4.35% | 115 | 2.46% | 1 | 2 | 2 | 2 | 3 |
| gr/gr dup | 149 | 1.29% | NA | NA | 3 | 4 | 6 | 2 | 5 |
| b2/b4 dup | 6 | 0.05% | NA | NA | 4 | 6 | 8 | 2 | 6 |
| gr/gr del + b2/b4 dup | 21 | 0.18% | NA | NA | 2 | 4 | 4 | 2 | 4 |
| b2/b3 del | 81 | 0.70% | 105 | 2.25% | 1 | 1 | 2 | 2 | 3 |
| b2/b3 or g1/g3 dup | 21 | 0.18% | NA | NA | 3 | 5 | 6 | 2 | 5 |
| unknown | 228 | 1.98% | NA | NA | NA | NA | NA | NA | NA |
| total nb of events | 1011 | 8.77% | NA | NA | NA | NA | NA | NA | NA |

* from Rozen et al. 2012.

**Table 2 | De novo mutation rate in patrilines for amplicons and palindromes**

| Amplicon | Type | amplification | deletion | total | mutation rate** (number of event per meiosis) | Percent of deletion (95% CI) |
|---|---|---|---|---|---|---|
| Yellow | MD | 13 | 4 | 17 | $1.98 \times 10^{-3}$ ($1.15 \times 10^{-3}$, $3.17 \times 10^{-3}$) | 23.5 (7.8, 50.2) |
| | SD | 16 | 24 | 40 | $2.53 \times 10^{-3}$ ($1.81 \times 10^{-3}$, $3.45 \times 10^{-3}$) | 60.0 (43.3, 74.7) |
| | tot | 29 | 28 | 57 | $2.34 \times 10^{-3}$ ($1.77 \times 10^{-3}$, $3.03 \times 10^{-3}$) | 49.1 (35.8 62.6) |
| Green | MD | 12 | 4 | 16 | $1.86 \times 10^{-3}$ ($1.07 \times 10^{-3}$, $3.03 \times 10^{-3}$) | 25.0 (8.3, 52.6) |
| | SD | 15 | 26 | 42* | $2.66 \times 10^{-3}$ ($1.92 \times 10^{-3}$, $3.59 \times 10^{-3}$) | 61.9 (45.6, 76.0) |
| | tot | 27 | 30 | 58* | $2.38 \times 10^{-3}$ ($1.81 \times 10^{-3}$, $3.07 \times 10^{-3}$) | 51.7 (38.3, 64.9) |
| Red | MD | 12 | 4 | 16 | $1.86 \times 10^{-3}$ ($1.07 \times 10^{-3}$, $3.03 \times 10^{-3}$) | 25.0 (8.3, 52.6) |
| | SD | 17 | 25 | 42 | $2.66 \times 10^{-3}$ ($1.92 \times 10^{-3}$, $3.59 \times 10^{-3}$) | 59.5 (43.3, 74.0) |
| | tot | 29 | 29 | 58 | $2.38 \times 10^{-3}$ ($1.81 \times 10^{-3}$, $3.07 \times 10^{-3}$) | 50.0 (37.5, 62.4) |
| Teal | SD | 1 | 1 | 2 | $1.27 \times 10^{-4}$ ($1.53\& \times 10^{-5}$, $4.57 \times 10^{-4}$) | - |
| | tot | 1 | 1 | 2 | $8.20 \times 10^{-5}$ ($9.93 \times 10^{-6}$, $2.96 \times 10^{-4}$) | - |
| P4 | SD | 1 | 0 | 1 | $6.33 \times 10^{-5}$ ($1.60 \times 10^{-6}$, $3.53 \times 10^{-4}$) | - |
| | tot | 1 | 0 | 1 | $4.10 \times 10^{-5}$ ($1.04 \times 10^{-6}$, $2.29 \times 10^{-4}$) | - |
| P5 | SD | 2 | 0 | 2 | $1.27 \times 10^{-4}$ ($1.53 \times 10^{-5}$, $4.57 \times 10^{-4}$) | - |
| | tot | 2 | 0 | 2 | $8.20 \times 10^{-5}$ ($9.93 \times 10^{-6}$, $2.96 \times 10^{-4}$) | - |
| P6 | SD | 1 | 0 | 1 | $6.33 \times 10^{-5}$ ($1.60 \times 10^{-6}$, $3.53 \times 10^{-4}$) | - |
| | tot | 1 | 0 | 1 | $4.10 \times 10^{-5}$ ($1.04 \times 10^{-6}$, $2.29 \times 10^{-4}$) | - |

* With one unknown event.

** estimated as the ratio between the number of events and the number of meiosis observed in the patrilines, with a 95% Binomial confidence interval. The total number of meiosis observed in the patrilines is 24380, with 8581 in branches with multiple descendants (MD) with observed CN genotypes and 15799 in branches with single descendants (SD) with observed CN genotypes.

affects only the green amplicon appears to be a partial duplication of this amplicon.

We detected several de novo events in the patrilines that result in the recovery of the ancestral (reference) state of 2 copies of a palindrome from a non-reference state. Thus, for palindrome 1, we observe 20 deletions from 3->2 copies and three duplications from 1->2 copies (Supplementary Fig. 9). Intriguingly, one of the latter rescue duplications was preceded 2 generations earlier in the same patriline by a deletion from 2->1 copies, showing how dynamic CN mutations can be in palindrome 1. The distribution of coverage along the palindromic arms suggests that individuals with a single arm of palindrome 1 have a gr/gr deletion that was followed by a b2/b3 duplication in individuals with recovery to 2 copies. This result was further validated by examining the MARF of individual PSVs across palindrome 1, which demonstrated a loss of pseudo-heterozygosity at paralogous positions in the yellow amplicon for individuals with 1 and 2 copies after recovery (Supplementary II.6, Supplementary Table 14). Overall, 21 individuals (0.18% of our sample) carry a CN of the green, red, yellow and teal amplicons that is consistent with a gr/gr deletion followed by a b2/b3 duplication (Fig. 1F, Supplementary Table 15).

## The long-term evolution of CN variants

It is of interest to determine whether evolutionary forces other than mutations have shaped the frequencies of CN genotypes observed in Icelanders. To this end, we estimated the steady state CN genotype frequencies expected from the matrix of mutation probabilities derived from the de novo events for reference and non-reference CN genotypes (Supplementary Table 16, and Supplementary section II.5). Using this as a Markov chain transition matrix, we determined the steady state frequencies of CNs assuming neutral evolution and infinite population size. We also constructed separate matrices and computed steady states for the low and high boundaries of the 95% CI of the mutation probabilities, respectively. The observed frequency of reference CN genotypes in Icelanders (0.914) is very different from the

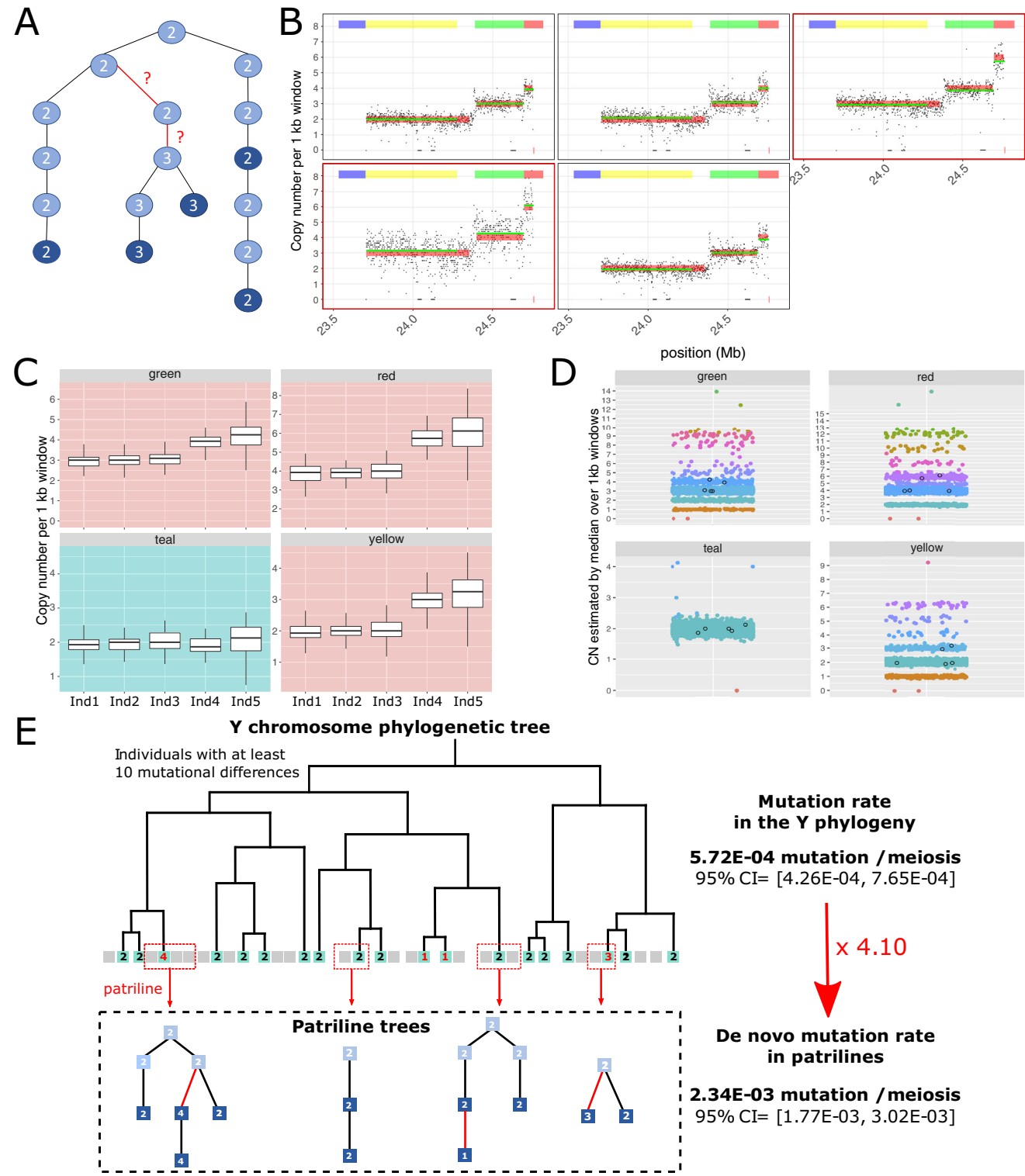

steady state expectation of 0.59 (95% CI = [0.28, 0.83]). The discrepancy between the observed proportion of Icelanders with reference CN and the steady state expectation could indicate that the pattern of mutations in the distant past was different from that observed in contemporary Icelanders or that non-reference CNs are somehow deleterious and subject to negative selection.

Assuming neutral evolution and a constant mutation rate across time, we expect estimates of the phylogenetic mutation rate to be close to those of the patrilineal mutation rate. The most important difference is that some recurrent mutations will be missed on long branches in a phylogenetic tree. To shed further light on the long-term mutation rate, we sought to estimate the phylogenetic mutation rate based on a phylogenetic tree constructed from 891 Icelandic Y chromosomes based on 23,118 bi-allelic sequence variants from the X-degenerate regions (8,974,361 positions). To avoid redundancy from closely related Y chromosomes and overlap with de novo mutations identified in the patrilines, the subset of 891 males had been pruned such that all pairs had at least 10 mutational differences at the X-degenerate sites. Applying the same parsimony approach used for mutation detection in the patrilines, the CN genotypes of individuals in

**Fig. 2 | Detecting de novo mutations in patrilines. A** An example of a patriline with a mutation event carried by multiple descendants (MD) leading to increased CN (2 -> 3) of the yellow amplicon. Each circle represents a male in the patriline, with lines representing the transmissions of Y chromosomes from father to son and the number inside each circle representing the observed (dark blue) or inferred (light blue) CN genotype. Red lines represent the transmissions where the mutation occurred. **B** For the five males with observed CN genotypes from the patriline shown in panel A, we show copy number across non-overlapping 1 kb windows for the yellow, green and red amplicons. The carriers of the mutation are indicated by a red frame. The position of each amplicon is indicated with coloured rectangles above the plots. **C** Boxplot representing the distribution of the CN across non-overlapping 1 kb windows within the green, red, teal and yellow amplicon, for each of the five males with observed CN genotypes in the patriline. Here we see that the 2 -> 3 increase in CN affects the green, red and yellow amplicons, but not the teal amplicon. **D** A summary of CN genotypes of all males with sequencing data, based on the median of the medians across non-overlapping 1 kb windows for each of the four amplicons. Each circle represents one male, with colours indicating the CN genotype called. The five sequenced members of the patriline in panel A are represented by black circles. **E** Comparison of the mutation rate estimated from the Y chromosome phylogenetic tree and the de novo mutation rate estimated using patrilines. A schematic representation of the Y chromosome phylogenetic tree (top) and four patrilines (bottom). The phylogenetic tree was constructed from individuals with at least 10 mutational differences according to haploid genotypes from the X-degenerate region of the Y chromosome, in order to minimise overlap between transmissions in the patrilines and the phylogenetic tree. As in the case of the patrilines, CN genotypes for ancestral males in the phylogenetic tree were inferred using parsimony, conditioning on the males with observed CN genotypes and their position in the tree.

### Table 3 | Amplicon mutation rates based on the phylogenetic tree

| Amplicon | Amplifications* | Deletions* | Phylogenetic mutation rate (95% CI)** (number of event per meiosis) | Percent of deletions (95% CI) |
|---|---|---|---|---|
| Yellow | 24 (2) | 24 (2) | $5.72 \times 10^{-4}$ ($4.26 \times 10^{-4}$, $7.65 \times 10^{-4}$) | 50 (36.4–63.6) |
| Green | 32 (5) | 23 (2) | $6.55 \times 10^{-4}$ ($4.98 \times 10^{-4}$, $8.60 \times 10^{-4}$) | 41.8 (28.9–55.9) |
| Red | 28 (4) | 22 (2) | $5.96 \times 10^{-4}$ ($4.47 \times 10^{-4}$, $7.92 \times 10^{-4}$) | 44 (30.3–58.7) |
| Teal | 1 (0) | 0 (0) | $1.19 \times 10^{-5}$ ($6.22 \times 10^{-7}$, $7.74 \times 10^{-5}$) | 0 (0.0–0.9) |

*the subset of mutations also detected in the patrilines are shown in parentheses.

**estimated as the ratio between the number of events and the number of meiosis observed in the phylogenetic tree (83,909), with the proportion test 95% CI.

the phylogenetic tree revealed 48 CN substitution events, each of which was assigned to a particular branch (Fig. 2E, Table 3). As for the patrilines, the substitutions typically affect multiple amplicons (Supplementary Table 17). Four events were also detected as de novo events in the patrilines (2 amplifications and 2 deletions), 32 are fixed in the patriline of the individuals used in the phylogenetic tree or detected in several individuals not included in patrilines, 11 are singletons and do not belong to a patriline (only observed in a single individual) and one is unclassified.

To estimate the CN phylogenetic mutation rate, we calibrated branch lengths in the phylogenetic tree using a SNP mutation rate of $3.07 \times 10^{-8}$ per position per generation, estimated for the Y chromosome X-degenerate regions in the Icelandic population[25]. This yielded a sum of branch lengths of 83,909 generations (95% CI = [75,766 − 93,333]) or 2,861,297 years (95% CI = [2,583,621–3,182,655]), based on 34.1 years per generation. The resulting phylogenetic mutation rate for the yellow amplicon was estimated to be $5.72 \times 10^{-4}$ per generation (proportion test 95% CI = [$4.26 \times 10^{-4}$, $7.65 \times 10^{-4}$]). This CN phylogenetic mutation rate is slightly higher than estimated by Teitz et al.[1] of $3.83 \times 10^{-4}$ per generation. However, that study used a slightly lower SNP mutation rate used to calibrate the time in their phylogenetic tree, namely $0.76 \times 10^{-9}$ mutations per bp per year (based on Fu et al. 2014). Our point estimate for the CN phylogenetic mutation rate based on an equivalent calibration is $4.83 \times 10^{-4}$ per generation.

### A direct assessment of selection on CN genotypes

To directly assess the impact of selection on CN genotypes in contemporary Icelanders, we obtained information about the number of children the males fathered from the deCODE Genetics genealogical database (birth year range 1900 to 2000, mean = 1956.1). To adjust for changes in the number of children over time in the population (Supplementary Fig. 10A and B), we grouped genotyped individuals into 5-year intervals based on birth year and performed a Z transformation of the number of children within each group (Supplementary Fig. 10C and D). Using a regression test, we did not observe a significant impact of non-reference CN (927 males) versus reference CN (10460 males) for the yellow amplicon, on the mean Z-transformed number of children (0.03 vs 0, p-value = 0.42, Supplementary Table 18). Similar results were obtained when deletions (N = 613) and duplications

(N = 314) were tested separately against reference CN. Neither did we observe a significant difference in the proportion of individuals with no children among the two genotype categories based on a Fisher Exact test (0.11 vs 0.12, p-value = 0.26, Supplementary Table 19). We note that most of the deletions (N = 495 out of 613) can be classified as gr/gr microdeletions, which have been reported in the literature to be associated with a low sperm count (oligozoospermia, at odds ratios ranging 1.76 to 2.4[15]), but we did not observe an impact on fertility.

These results rule out the action of strong negative selection against non-reference genotypes during recent generations in Iceland. However, it is possible that weak selection gradually removes deleterious variants from the population, but at a rate not easily detectable in the few generations provided by our set of contemporary Icelandic males. We, therefore, sought to determine the magnitude of negative selection against non-reference CN genotypes that would be needed to yield the observed frequencies in the Icelandic population, given the two-state mutation matrix (reference vs non-reference CN) derived from the patriline de novo events – in other words, the magnitude of negative selection to balance the observed and steady-state frequencies. To this end, we simulated a population where everyone in the first generation has the reference CN for the yellow amplicon and subsequent generations give rise to new variants in accordance with the de novo mutation matrix (Supplementary Table 11, Supplementary section IV.4). Testing a range (0 to 0.05) of selection coefficients (s) against non-reference CN genotypes, we found that s = 0.018 (1.8% negative selection) was most likely to produce the observed CN frequencies in the Icelandic population (Supplementary section IV.4, Supplementary Fig. 11-12). Given a sample size of 11,527 men (10,460 with the reference CN), a mean of 2.44 children and s = 0.018, we estimate that the power to detect a difference in the number of children using a t-test is only 16% (Supplementary section IV.5, Supplementary Fig. 13). Moreover, under these conditions and assuming the number of children is drawn from a Poisson distribution, with a mean of 2.44 for 10460 men with reference CN and 2.44 * (1-s) for 927 men with non-reference CN, there is a 23% chance that men with non-reference CN have at least as many children as those with reference CN, as observed in our data. We therefore conclude that the lack of association between CN genotype and the number of

children in our data is not inconsistent with a selection coefficient of 1.8% acting against non-reference CN.

Some non-reference CN variants, such as CN = 1 in haplogroup N[26], are known to be at least several thousands of years old, and therefore unlikely to be subject to negative selection. We thus considered the possibility that CN variants deemed identical by our genotype calls, could in fact have different impact on fitness, due to some sequence difference that is not captured by our method of CN genotype-calling. In this situation, one would expect an enrichment of deleterious variants among those that are very recent, such as variants derived from the de novo mutations detected in patrilines. We did not observe a significant difference in the number of children between the 91 individuals carrying de novo mutations and the 10460 individuals with the reference CN (t test $p$-value = $8.81 \times 10^{-2}$). However, the power to detect such a difference, assuming s = 0.018 is only 5.9% (Supplementary section IV.5, Supplementary Fig. 13C).

Finally, to gain a broader view of the phenotypic impact of CN variation in the yellow amplicon, we performed association tests against a diverse set of 341 case-control (number of cases ranging from 11 to 2808, with all other CN genotyped men as controls) and quantitative traits (N ranging from 562 to 10577), including widely tested common diseases, anthropometric traits and blood measurements (Supplementary Data 2-4). None of the association tests were statistically significant after adjustment for the number of tests performed. As the $gr/gr$ deletion has been reported to be associated with risk of testicular cancer[27], we specifically tested for association against our CN genotypes. Our set of males only included 51 males with a diagnosis of testicular cancer. Among those 51 males, 47 have the reference CN for the yellow amplicon, we therefore do not have statistical power to detect weak associations between copy number variations and testicular cancer. For the $gr/gr$ deletion, we observe an odds ratio of 0.445 (95% CI 0.011 − 2.612), which at least makes it unlikely that there is a strong association between this variant and testicular cancer. Overall, we conclude that the phenotypic impact of the Y ampliconic CN variation is somewhat limited in humans.

## Discussion

Many previous studies have reported an effect of large *AZFc* deletions on fertility phenotypes, from mild oligozoospermia to complete azoospermia[28]. For smaller CN variants, such as *gr/gr* deletions, the results are inconsistent between studies[15,18,19,29]. Some studies have claimed a differential impact of *gr/gr* deletions by ethnicity[18,29], while others have proposed differential impact by haplogroup[17]. A recent meta-analysis[19], based on data from multiple studies of men from different ethnicities (10,978 Y chromosomes from infertile men and 6704 from fertile/normozoospermic controls), reported an odds ratio of being infertile of 1.8 for men carrying a *gr/gr* deletion. Specifically, in European males, *gr/gr* deletions are observed in 7% of infertile against 2% of fertile men, suggesting some contribution to infertility[19]. In addition to deletions and duplications, there are also more complex structural variations like inversions that could have an impact on phenotypes[30], but are not detectable using the approach employed in this study.

Our study of CN variation in the *AZFc* region of 11,527 men from the Icelandic population, shows no direct evidence of impact of non-reference CN on the number of children or 341 other GWAS phenotypes. However, we do find indirect evidence of weak negative selection, of a magnitude close to 1.8%, based on the discrepancy between the rate of de novo mutations and the current frequency of CN variants in Icelandic men and the much higher rate of mutations in patrilines than in the phylogenetic tree.

It is also possible that some relaxation of negative selection against non-reference CN genotypes has occurred during the last century due to the introduction of assisted reproduction. However, while assisted reproduction would be expected to counteract a deficit

in fertility in some cases, it cannot have done so completely, as many of the men in our cohort were reproductively active before such treatment options became available in Iceland or elsewhere in the 1980s. We note that the measurement of additional fertility-related phenotypes, such as sperm counts or sperm motility might shed more light on the impact of non-reference CN genotypes and could reveal direct effects on spermatogenesis. However, it is intriguing that copy number variation on the Y chromosome has so little impact on phenotype variation among men.

## Methods

A detailed description of all analyses carried in this study out is included in the Supplementary Methods.

### Inclusion and ethics statement

The study was undertaken on the basis of approvals from the National Bioethics Committee and the Icelandic Data Protection Authority. Blood or buccal samples were taken from individuals participating in various studies, after receiving informed consent from them or their guardians.

### Copy number estimations

We estimated copy number of 11,527 males from Iceland, part of Decode sequencing dataset, including 7947 males belonging to 1449 patrilines. All individuals were sequenced using PCR-free method at a mean coverage of 30X. Haplogroup assignment was performed using HaploGrouper[31] and the 2016 version of the ISOGG Y chromosome phylogenetic tree. We constructed a reduced Y chromosome fasta sequence composed of the proximal arm sequences for palindrome 1, 3, 4, 5, 6, 7 and 8 of the Y chromosome, in which all annotated genes were masked. The Y-linked single copy genes (AMELY, DBY, EIF1AY, KDM5D, NLGN4Y, PCDH11Y, PRKY, RPS4Y1, RPS4Y2, SRY, TBL1Y, TGIF2LY, TMSB4Y, TXLNGY, USP9Y, UTY and ZFY) and one copy of each ampliconic gene sequence (+− 2 kb upstream and downstream) were then added to the final Y chromosome mapping file (see Supplementary Tables 1–3) for the hg38 positions of the sequences included). The pipeline described in Lucotte et al. 2018 and Skov et al. 2017 was used for determining the copy number of genes. The median coverage of all Y single copy genes was used to obtain a measure of the expected coverage for single copy for each individual.

We also inferred copy number along the amplicons and palindromes using a hidden Markov model with copy numbers as hidden states. The HMM assigns copy number based on the median coverage of non-overlapping sliding windows and uses the median coverage of all single copy gene as a reference for the 'one copy' state. The probability of staying in the same state was set to 0.999, and the distribution of the HMM was a Poisson distribution, with a parameter defined as the median coverage of all single-copy genes. For the amplicons located in P1, P2 and P3, windows of 5000 bp were used in the HMM, while windows of 1000 bp were used for the other palindromes because of their smaller size. Unlike the previously described method, the HMM yields an integer estimate, allows for changes in copy number within a region and can detect duplication and deletion breakpoints. Windows with more than 10% of missing data were not considered. For yellow, green and teal amplicons and P4 and P5, several sub-regions were removed from the HMM analyses due to low coverage, because the genes were masked in the reference sequence or because of poor mapping (Supplementary Tables 19–21).

Finally, we estimated the copy number of amplicons and palindromes over non-overlapping windows of 1 kb. For each window, the median coverage was computed and divided by the median coverage of single copy genes. The median of the median coverage of each window was then computed.

### Filtering for ambiguous copy number

We defined a copy number estimate as ambiguous if one of the following applied: *i)* a breakpoint was detected by the HMM but not validated, *ii)* a difference >=0.5 exists between the HMM CN estimate and the one based on the median of the medians per window and *iii)* the CN estimate is located outside of their cluster CN + −0.35 (see Supplementary Methods for further details on the breakpoint validation protocol and the cluster definition). Individuals with ambiguous copy numbers were not removed from further analyses, and their ambiguous CN was considered for the detection of events in patrilines and in the Y chromosome phylogenetic tree.

### Assessing the accuracy of the CN estimates using duplicates

The availability of individuals which have been sequenced independently more than once (duplicates) allowed us to investigate the robustness of our copy number estimates. Two sets of duplicated sequencing were available: from different sample types (blood and buccal, 77 individuals) and from the same sample type (both blood or both buccal but from samples taken at different times, 14 individuals). The same pipeline to assess CN was applied to the duplicates, and the CN estimates were compared for each pair of duplicates. Individuals with ambiguous CN were not considered for this analysis. We report the differences in CN estimated in Supplementary Methods section 1.5.a, Supplementary Tables 22 and 23).

### Validation methods based on independent mapping

For a subset of 9162 individuals, mapped independently to the entire reference genome (hg38)[24] we performed two validations analyses. The software lastz[32] was used to identify paralogous sequences in the yellow amplicon. Using this information, we combined all reads for each set of paralogous positions per individual. We next identified the subset of combined paralogous positions with evidence of allelic variation, so-called paralogous sequence variants (PSV). For the PSVs, we calculated the mean minor allele read frequency (MARF) for each individual, which can be informative about copy number.

We used only PSVs, where at least one male had MARF > 0. A number of filters were implemented per PSV in order to exclude false positives (see Supplementary Methods for details). These filters result in 4139 PSVs for palindrome 1 and 2334 PSVs for the yellow amplicon.

We performed a second validation analysis using relative sequence depth (rdepth) to infer CN. The software samtools bedcov was used to calculate the average number of reads mapped to 8,974,361 positions in the Y chromosome X-degenerate regions for each individual (see Supplementary Table 24 for the exact coordinates used) · yielding an average across all individuals of 19.51 (sd = 7.26, median = 18.8). For the combined paralogous positions in the yellow amplicon, identified using lastz (described above), we obtained the read depth per paralogous position set for each male for yellow 1 and yellow 2. Division by the average per-position X degenerate read depth was used to calculate rdepth for each paralogous position set in each male, which was in turn used to estimate CN in 1000 bp windows across the yellow amplicon.

### Mutation rate within patrilines

The patrilines constructed using genealogies were corrected using SNP genotype data from the X-degenerate region of the Y chromosome (as described in ref. 25, see Supplementary Methods for more details). We used a parsimony algorithm, as described in[33], to assign genotype states to ungenotyped ancestors in the patriline trees and thereby to assign mutation events in cases where genotype variation was observed among sequenced patriline members. In cases where the ancestral state of a patriline could not be inferred using the tree, we defined the ancestral state as the reference CN if one of the leaf CN states was the reference. Otherwise, we checked which was the most probable ancestral state using the tree and the distribution of CN in the

haplogroup of the individuals. If all else failed, we assigned an unknown CN as ancestral state, and the direction of the event could not be determined.

Each event was categorised as Multiple Descendent (MD) if the mutation had several carriers or Single Descendent (SD) if the mutation had one carrier. It was further determined whether an event was an amplification or a deletion, if the ancestral state had been determined. The events were manually curated for individuals that were assigned an ambiguous CN. If an individual with an ambiguous CN in one of the amplicons was carrier of an SD event in another amplicon and if the ambiguity of the CN in this individual removed the event, we removed it from further analyses. See Supplementary Methods section II for more details on mutation rate estimations in patrilines, including validations, mutation probability matrix construction and the analysis of a rescue event.

### Phylogenetic mutation rate in the Y chromosome phylogenetic tree

To construct a phylogenetic tree based on the Y chromosomes of the Icelandic males used in study, we first identified a set of good quality polymorphic SNP variants from the X-degenerate regions (8,974,361 positions). Specifically, we called genotypes for all X-degenerate positions using the approach described in ref. 25 and sought SNPs that were variable among the set of 11,527 males analysed in this study. The SNPs selected ($N = 39,001$) were those where all alleles were supported by >5 reads in total and >3 reads on average and where 99% of the males were assigned unambiguous haploid genotypes.

In order to minimise the overlap between CN mutation events detected in the phylogenetic tree and the patrilines and to make the construction of the phylogenetic tree more tractable, we pruned the full set of males down to a subset, such that no pair of males was separated by fewer than 10 genotype differences at SNPs. This led to the removal of 10,636 males, leaving 891 males to be used for constructing the phylogenetic tree. We used the neighbour-joining algorithm, as implemented in RapidNJ[34], to construct the tree from pseudo-sequences derived from the haploid genotypes of the 891 males at 23,118 variable SNPs. To obtain branch length estimates, the SNPs were assigned to the branches of the neighbour-joining tree (using a parsimony algorithm) and then the SNP mutation rate for the X-degenerate regions[25] was used to calibrate the branch lengths to meiosis, yielding a total of 83,909 meiosis in the tree (95% CI 75,765.9–93,333.5, based on the SNP mutation rate confidence intervals) and using 34.1 years per meiosis as a conversion factor to years. The branch structure of the neighbour-joining tree was then used to detect CN mutation events.

The same parsimony algorithm was used to infer ancestral states and mutation events in the phylogenetic tree. The ancestral state of the phylogenetic tree was set as the reference CN.

The phylogenetic mutation rate was estimated by dividing the number of events by the number of meiosis included in the tree. The number of meiosis in the phylogenetic tree is 108639.05. Phylogenetic mutation rates were calculated for each amplicon. The confidence intervals were estimated using a proportion test. Events were also grouped into independent set of events to obtain the phylogenetic mutation rate in the *AZFc* region (Supplementary Table 16).

### Phenotypic associations

First, we considered two phenotypes that are proxies of fertility: the proportion of individuals with no children and the mean number of children. We compared these values between groups of individuals with different CN, using a Fisher exact test and a Wilcoxon Mann Whitney test, respectively. We applied two filters on birth year, first for individuals born before 1980 and second for individuals born before 1970. Categories including less than 15 individuals were not tested.

A diverse and curated set of 342 quantitative phenotypes was used to test association with CNV.

We compared non-reference CN with reference CN, deletion with reference CN and duplication with reference CN. Logistic regressions were performed for binary phenotypes, and linear regression were performed for quantitative phenotypes.

**Simulations.** We performed simulations to determine the level of selection (parameter s) acting on non-reference CN that would be necessary to explain the observed frequencies of non-reference CN in Iceland, considering the mutation matrix. See Supplementary Methods section IV.3 and Supplementary Table 25 for details on the simulations.

**Power analyses.** We determined the strength of selection necessary to detect a significant difference between the number of children of individuals carrying the reference CN and individuals carrying a non-reference CN. Details are available in Supplementary Methods section IV.4.

### Reporting summary
Further information on research design is available in the Nature Portfolio Reporting Summary linked to this article.

## Data availability
Icelandic law and the regulations of the Icelandic Data Protection Authority prohibit the release of individual level and personally identifying data. We are actively participating in multiple meta-analysis based on our data and are in collaboration with groups at over 100 international universities and institutions. Therefore, collaborations based on our sequencing data are based on the release of summary level statistics, such as effect sizes and P-values for meta-analysis, or the collaborators travelling to our Icelandic facilities for local data access. Contact the corresponding author Agnar Helgason (agnar.helgason@decode.is) for more details.

## Code availability
The python codes used for the simulations are available upon request.

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

## Acknowledgements

The study was supported by grant NNF18OC0031004 from the Novo Nordisk Foundation and grant 6108-00385 from the Research Council of Independent Research. VBG was supported by a PhD grant from the University of Iceland Research Fund.

## Author contributions

E.A.L., V.B.G., J.M.J. and A.H. analysed the data with input from L.S., M.C.M., K.A., A.H. and M.H.S. E.A.L., J.L.J., A.H., K.S. and M.H.S. designed the study. E.A.L., V.B.G., A.H. and M.H.S. wrote the manuscript with input from all authors.

## Competing interests

V.B.G., A.H. and K.S. are employees of deCODE Genetics and Amgen. The remaining authors declare no competing interests.
