## [Peer Review File · Nature Communications]

Characterizing the evolution and phenotypic impact of ampliconic Y chromosome regionsREVIEWER COMMENTS

Reviewer #1 (Remarks to the Author):

This paper studies the evolution of Y-chromosome amplicons and could be a sequel to Repping et al. 2006, Nat Genet 38:463 and Teitz et al. 2018, Am J Hum Genet 103:261. The authors have access to an unparalleled and powerful dataset, including more than 11,000 male genomes sequenced to 30X coverage (15X for the Y). However, the authors neglect to indicate which sequencing platform was used (e.g., Illumina), which should be clarified. The number of samples is an order of magnitude greater than in Teitz et al., 2018, which included samples representing Y chromosomes from entire human population. This current study includes samples from one small population, Iceland, and many of the subjects are closely related, which allows the investigation of mutational processes at a much finer scale. The results confirm the conclusions of Teitz et al., 2018, as the great majority of Y chromosomes have the same amplicon copy numbers as the reference Y chromosome, presumably as the result of selection against amplicon copy number changes.

Unfortunately, the authors' knowledge of the relevant human Y chromosome literature appears to be quite limited and superficial. The authors need to become better acquainted with the relevant literature and reframe the paper accordingly. I list below numerous specific examples:

- Ln. 45: "In light of [the Y-chromosome's degeneration], it seems surprising that many of the genes on the human Y chromosome are essential for male fertility (Krausz and Casamonti 2017)." There is no existing literature on the Y chromosome, dating back decades, that indicates that the presence of male fertility genes on the Y chromosome would be unexpected in some way. In fact, quite the opposite is true.
- Ln. 47: Ampliconic genes should be described as "testis-specific or testis-biased" instead of "testis-expressed." Most genes in the genome are expressed in testes (as well as in other tissues).
- Ln. 50: "The male-specific Y (MSY) region contains eight key palindromes, with a well-established numerical nomenclature: P1 to P8 (Krausz and Casamonti 2017)." The reference for palindrome nomenclature should be Skaletsky et al. 2003, Nature 423:825.
- There is an additional misrepresentation of the history of MSY nomenclature in Supplementary Information, pg. 3: "We classified amplicon CN genotypes using a system of nomenclature devised to describe recurrent deletions and duplications in the AZFc region of the Y chromosome (reviewed in (NavarroCosta et al. 2010), Figure 1F). This nomenclature provides information about the location of

deletion or duplication breakpoints, by listing the first letter of the amplicon colour label (g for green, r for red, etc. because it was originally detected with coloured probes using FISH.)” The reference for this nomenclature should be Kuroda-Kawaguchi et al. 2001, Nature Genetics 29:279. The color names assigned to amplicons do not correspond to FISH probes.

- Ln. 56: “The inverted-repeat structure of the palindromes promotes pairing between their arms, which provides a mechanism for the rescue of deleterious mutations in the ampliconic genes through gene conversion (Rozen et al. 2003; Hallast et al. 2013; Skov et al. 2017). This pseudo-diploid pairing of palindrome arms can also result in structural mutations, such as deletions and duplications, which are recurrent in human populations (Skov et al. 2017; Lucotte et al. 2018; Teitz et al. 2018; Ye et al. 2018), especially in the AZFc region.” This statement shows a lack of understanding of the mutational process under study. Deletions and duplications arise from ectopic recombination between direct repeats on the Y chromosome. Intrachromatid pairing/recombination between opposing palindrome arms, which are inverted repeats, result in typically harmless inversions. However, interchromatid recombination between opposing palindrome arms can result in deleterious isodicentric chromosome formation (Lange et al. 2009, Cell 138:855). The authors should clarify that recurrent deletions on the Y chromosome arise from recombination between direct repeats, i.e., distinct copies of the same palindrome.

- Ln. 103: “The grey amplicon is located outside P1 and was omitted from our analysis.” The gray amplicon is actually part of P1.

- Ln. 104: “The blue amplicon (167,703 bp) and P7 (23,149 bp) were omitted because their CN genotypes could not be called accurately, primarily due to relatively small size.” Only b1, which is an incomplete blue amplicon, is 168 kb; the other three blue amplicons are 230 kb. In Teitz et al. 2018, the number of blue amplicons was successfully defined.

- This study does not include (or even mention) the one non-reproductive phenotype associated with AZFc deletions – testicular cancer (Nathanson et al. 2005, Am J Hum Genet 77:1034). This study includes a large group of samples with the gr/gr deletion, which has been linked to testis cancer, presenting an excellent opportunity to investigate the frequency of testis cancer in these men and potentially corroborate the existing link.

This paper also has serious design flaws and omissions, which are listed below:

- A critical aspect of a study measuring copy number variation is the choice of single-copy sequence for control. In Teitz et al. 2018, the control sequence was selected carefully, comprising several long stretches of Y-specific sequence from the parts of the Y unlikely to be deleted or duplicated. These intervals also served as controls for each other, so that if any one of them was deleted in some samples,

it would not skew all the counts. This design could also be used in this study, but instead the authors come up with a scheme focused on the genomic sequence of single-copy Y genes as a control. The problem is that this equates to numerous short intervals, many of which can be deleted or duplicated. They also do not represent absolutely Y-specific sequence since they have homologs on the X chromosome. The inclusion of TGF2LY and PCDH11Y from the Y chromosome's X-transposed region, which is 99% identical to the X, almost guarantees mismapping of some reads.

- The data in this study (as in Teitz et al. 2018) allows only the estimation of amplicon copy number; inferences about amplicon arrangements are speculative. Therefore, all conclusions should be based solely on amplicon numbers. Some of the known recurrent rearrangements do remove or duplicate whole palindromes, but most do not, and this kind of analysis is not capable of distinguishing between whole palindromes and multiple copies of single palindrome arms. Nonetheless, the authors discuss rearrangements in terms of palindrome number (e.g., ln. 198-199 and Table 2 in the main text; sections 5 and 8 in Supplementary Information).
- The authors validate their method by analyzing individuals closely related to the individuals in the study, or by using independently obtained sequence from the same individuals. They get very similar results and consider this proof of the validity of the method. However, this only proves consistency. To validate their method, they would need to use an independent method to estimate copy number in samples with variations, such as FISH or quantitative PCR. If additional experiments cannot be done, the authors could process some samples from Teitz et al. 2018 through their pipeline and compare the results. Unlike the raw data from this study, the data sets from Teitz et al. 2018 are identified and publicly available.
- Ln. 251: The authors' discussion of the CN substitution rate is confusing. They need to clarify whether they are talking about copy number variation of amplicons or substitution rate within amplicons.
- There is no information included about the data used in this study aside from the source: the deCODE Genetics genealogical database. Useful information to include would be sample IDs and criteria for selecting subjects (e.g., are they all of the men in their database, or only a subset?).
- It is understandable that the raw data may be not available for privacy reasons, but the primary results are not even provided. The authors should include coverage estimates and copy number calls, including amplicons and control intervals, for all samples.
- Repping et al. 2006 includes a list of predicted AZFc rearrangements that can occur starting from the reference organization through one, two, or three deletions/duplications (via ectopic recombination between direct repeats) or inversions (via ectopic recombination between inverted repeats). In both

Repping et al. 2006 and Teitz et al. 2018, each non-reference set of amplicon numbers was compared to those predicted rearrangements, and if there were matches, the one requiring the least number of steps was chosen as the likely explanation. The authors did not perform such a comparison in this study, so many of their “non-explained” cases could in fact be explained by the predictions in Repping et al. 2006.

- Ln. 352: “Measurement of additional fertility-related phenotypes, such as sperm counts or sperm motility might shed more light on the impact of non-reference CN genotypes and could reveal direct effects on spermatogenesis.” This statement implies that such measurements have never been done, while in fact numerous published studies have looked at these parameters in relation to AZFc deletions (e.g. Reijo et al. 1996, *Lancet* 347:1290; Visser et al. 2009, *Hum Reprod* 24:2667). This study used number of children as a proxy for fertility status/spermatogenic output, which is a poor substitute. Their results are summarized in Table S11, where the following mutations are included: gr/gr deletion, gr/gr deletion + b2/b4 duplication, b2/b3 or g1/g3 duplication, gr/gr duplication, b2/b3 deletion, b1/b3 deletion, b2/b4 duplication, and AZFc. The “AZFc” rearrangement is not mentioned in the text, so one can assume that it stands for the full AZFc deletion, which is also known as b2/b4. They found two such deletions in their population, and the mean number of children for these men is 2, which is only slightly lower than the mean number of children for men with an intact Y: 2.44. This finding confirms the fallacy of this substitute: It is well known that AZFc deletions cause severe oligozoospermia or azoospermia, and men with AZFc deletions who father children without the use of assisted reproduction are rare enough to warrant the publication of case reports (e.g. Chang et al. 1999, *Hum Reprod* 14:2689; Gatta et al 2002, *J Med Genet* 39:E27).

- There are three large spreadsheets showing absence of correlation between AZFc region rearrangements and various non-reproductive diseases. Apparently, this is considered one of the most important conclusions of the study, hence the final sentence: “Finally, it is intriguing that sequence variation on the Y chromosome, which on one hand contains the SRY gene being responsible for sex determination, and therefore the greatest normal differences in phenotypes among humans (and other mammals), otherwise has so little impact on phenotype variation among men.” However, they neglect to consider the impact of the Y chromosome’s single-copy genes, many of which are expressed across the body and likely have phenotypic consequences when deleted or mutated. There are recurring deletions on the Y – P5(P4)/P1, AZFa, and AMELY – that remove single-copy genes, which may be associated with non-reproductive phenotypes, yet it seems that the authors have not examined this possibility.

Other minor errors (likely not a complete list):

- Fig. 1A: The ampliconic gene family HSFY, located in P4, is not shown on the lower image of the chromosome.

- Supp. table S1B: The gene name DBY should be updated to DDX3Y. The location of HSFY is incorrectly indicated as P5.

Reviewer #2 (Remarks to the Author):

The manuscript by Lucotte and colleagues, "Characterizing the evolution and phenotypic impact of ampliconic Y chromosome regions", investigates the genetic diversity of the human Y-chromosomal palindromes. They have used whole-genome sequence data from a total of 11,527 Icelandic men to estimate the copy number (CN) changes in these regions, making this by far the largest study to date on the topic. They report the highest number of CN changes in AZFc region as is also previously known. Due to the nature of the data, they are able to estimate and compare the CN mutation rates derived from patrilineages and Y phylogenetic tree, and due to the obtained difference suggest that selection must be favouring the reference copy number. I think it is very important to improve our understanding about these complex Y-chromosomal regions and whether or not they affect any phenotypic traits.

Overall, I find this study to be of great interest to the scientific community. The Icelandic dataset is a great resource in addition to all else also for Y-chromosomal studies due to extensive pedigree information. I have some major comments to the authors, but assuming they can adequately address these, I believe it merits publication in Nature Communications.

Major comments:

1. I suggest the authors to try and re-organise the Results section so that it would have a more "logical" flow and be easier for the reader to follow, perhaps also consider shortening it (e.g. by moving some details to the methods section). Even though I very much appreciate the amount of work that has gone into producing this manuscript, it is not easy to read as it stands.

For example, the first section of Results, "Population variation", second paragraph on rows 100-105, starts describing the AZFc region straight away. I would suggest the authors first give a more general description of what was found in all the studied samples across all palindromes, how many rearrangements fit with the previously known deletion/CN events and how many are novel etc, and then focus in more detail on the most variable AZFc region. The same applies for other Results sections.

Also, for the description on rows 100-105 please specify that it refers only to AZFc region. Table 1 - please add the data from Teitz et al 2018 as it is currently the most thorough study of the Y-chromosomal ampliconic regions.

2. I have some concerns about the estimation of the phylogenetic mutation rate. Perhaps the authors could explain in more detail e.g. why specific samples were chosen for the phylogenetic tree

reconstruction, why 6 (or 10, see below) mutations were chosen to separate the samples, and whether the two approaches (phylogenetic vs patrilineages) produce mutation rates that can be directly compared or if some aspects should be considered while doing so, etc.

Also, there are confusing discrepancies between the description in the Supplementary and main text:

In the main text, lines 237-242: "To shed further light on the long-term mutation rate, we sought to estimate the phylogenetic substitution rate based on a phylogenetic tree constructed from 1405 Icelandic Y chromosomes based on 32,914 bi-allelic sequence variants from the X-degenerate regions (8,964,929 positions). To avoid redundancy from closely related Y chromosomes and overlap with 10 de novo mutations identified in the patrilineages, the subset of 1405 individuals used to produce the tree was pruned such that all pairs had at least 10 mutational differences at the X-degenerate sites."

Copy from the relevant section in Supplementary, page 10: "Two separate trees were constructed, and each was pruned such that no pair of males included in the tree has fewer than 6 mutational differences. The tree includes 2221 individuals."

3. Due to the large sample size of the study, the authors are in a great position to evaluate lineage-specific differences in CN rates and I believe this would substantially strengthen the paper. According to Figure S2, a number of major lineages (I1a1, I1a2, Q1a2, R1a1, R1b1) appear to have a good representation and include quite a few samples with identified CN changes. It would be very interesting to see if there are substantial lineage-specific differences e.g. in mutation rates, types rearranged regions etc. Perhaps the large number of samples with increased copy numbers result from a few sub-lineages where such copy numbers are fixed (e.g. I1a2 a lot of samples carry CN increases in the AZFc regions)?

Following from this, a recent paper by Hallast et al (DOI: 10.7554/eLife.65420) proposes an inversion in the AZFc region specific to haplogroup R1a1-M458 which makes these Y chromosomes more prone to subsequent deletions (e.g. *gr/gr* and *b2/b3*). It would be great to see if this can be confirmed in the current dataset. The authors can identify the carriers of R1a1-M458 chromosomes from the whole-genome sequence data and investigate if indeed these show increased levels of CN changes.

4. Curiously, the Hallast et al study reported that close to 20% of samples with *gr/gr* and *b2/b3* (mostly in haplogroup N) deletions had undergone secondary *b2/b4* duplications, whereas the current study reports only approx. 4.2% of samples with *gr/gr* deletion plus a secondary *b2/b4* duplication (21 out of 495, Table S11). Also, even though the number of haplogroup N1c1 samples (carrying the *b2/b3* deletion) does not appear very large in the current study, none of these seem to carry secondary duplications. Perhaps the authors can comment on this discrepancy.

5. Please add a table with the exact coordinates of X-degenerate regions which were used to reconstruct the Y-phylogeny. Also, please provide details about calling and filtering of these regions and about how the Y phylogeny was reconstructed.

Minor comments:

- row 45: "many of the gene" should be "many of the genes"

- row 71-72, sentence: "A change of CN within palindromes typically leads to a change of CN of ampliconic genes, which could have a phenotypic impact."

I find this sentence slightly misleading. Perhaps instead of "typically" it would be better to say that "can lead to a change of copy number", since palindromes P6 and P7 do not contain genes and also partial rearrangements within palindromes can happen which do not necessarily affect the genes.

- row 72, sentence: "Large deletions of parts of the Y chromosome are thought to cause infertility...". I would suggest changing "are thought to cause" to "are known causes of" or something similar as I do not think there is doubt about the complete AZF (a, b, c) deletions causing male infertility.

- Table S7 - please add phylogenetic marker names to the haplogroup label.

- Supplementary, page 3: 3) "Construction of the artificial Y chromosome - We first constructed a reduced Y chromosome fasta sequence composed of the proximal arm sequences for palindrome 1, 3, 4, 5, 6 and 7 of the Y chromosome, in which all annotated genes were masked."

Why was P8 excluded?

Reviewer #3 (Remarks to the Author):

This study leverages the spectacular sample sizes and genomic power of Decode, and therefore presents a very impressive dataset, in which whole-genome sequence data are used to investigate copy number variation of the Y chromosome (focusing on the ampliconic regions on Yq) in many thousands of Icelandic males whose genealogical connections are known.

The questions addressed – the rates and selective/phenotypic effects of Y-chromosomal CNVs – have been much debated over many decades and are clearly in need of further illumination through population resequencing efforts, which this study offers.

The study has taken a serious approach to the topic and presents much interesting data. However, there are several aspects that need further explanation and clarification.

1) The study addresses CNVs by normalized read-depth analysis, and uses known rearrangements of the AZFc region as an interpretative framework for this complex part of the chromosome. There are some issues here:

a) The methodology employed here is blind to inversions, which have previously been demonstrated to be important because they can generate ampliconic arrangements that are permissive for particular kinds of deletions and duplications. This fact should be stressed more clearly than it is, and some discussion of how such copy-number neutral rearrangements can be detected, and what their impact is, would be welcome. Currently quite a lot has to be assumed from the data presented here about the configuration of amplicons.

b) In the AZFc region, read-depth-based CNV analysis is used to assay copy number of the yellow, green, red, teal and grey amplicons, but not the blue. The text claims that the absence of the blue repeat is because of its small size (l.105), but this does not seem a reasonable explanation, because at 168 kb it is bigger than the red repeat (127 kb) or P8 (successfully typed at only 42 kb/arm). It seems more likely that failure to type the blue repeat is due to the distribution of copy numbers (Figure S2), with single unit differences between classes much commoner than for e.g. the red repeats (mostly two unit differences). It could also be due to differences in mappability of reads between different amplicon types, I guess. Whatever the reason, the absence of CNV data for blue is a nuisance. There are three of these units in the reference sequence, and they are involved in a lot of the NAHR-driven rearrangements of the region, including the classical AZFc deletion. I would urge the authors to reconsider the data for this repeat unit – the clustering in Figure S2 looks like it could be callable; if they really cannot include it, they need to do more to convince the reader that its absence does not affect their interpretation and conclusions. For example, could some of the 228 men with non-reference CN variants (l. 147-8) be interpreted if blue repeat data were available?

c) There is a recent study (doi: 10.7554/eLife.65420) of a rearrangement in the AZFc region that is not included in the authors' set, and perhaps should be.

2) l. 42-45: The beginning of the Introduction sets out an unconvincing argument to explain the rapid evolution of mammalian Y chromosomes – 'driven by degeneration because the Y chromosome cannot purge itself of deleterious mutations by means of recombination, with the exception of the small PARs'. Degeneration explains the loss of genes compared to the X, but is it not more likely that the lack of any pairing constraint in the MSY is responsible for toleration of rearrangements and thus the rapid evolution? In the relatively conserved PARs pairing is necessary. If the inexorable degeneration (aka Muller's ratchet) leads to deleterious mutations these are most likely due to be subject to negative selection and lost – some of the AZF deletions are examples of this.

3) Y chromosomes are assigned to haplogroups in the study, but I could not find anywhere a description of how this was done. Presumably SNPs were called from the sequence data and correlated to haplogroup nomenclature, but which SNPs, and which nomenclature? Some information is needed, and maybe a phylogenetic tree figure. Connected to this issue is haplogroup resolution. How were decisions made on how to split haplogroups when considering CNV? Figure S2, for example shows some haplogroups highly divided but others, e.g. R1b1, not. In this same figure, haplogroup names need more explanation. Presumably e.g. I1a means I1a* (excluding I1a1, I1a2, I1a3)? Please clarify. Also, define 'NA'.

4) Based on Figure S2, haplogroup I1a2 shows markedly high copy numbers compared to most other lineages. This deserves some comment.

5) The authors used pseudohomo/heterozygosity of variants within the yellow repeat to support their CNV calls. This is a good idea, but needs a bit more explanation for the reader. Why yellow? Is this because it's the biggest repeat unit and therefore has most variants? These variants are referred to as 'SNPs' (I.214 and several other places in the ms), but this term is also used for variants that are used to define lineages, and this will be confusing. Can another term be used to avoid this? Some other authors have used 'PSV' (paralogous sequence variant), so that could be considered.

6) It seems likely that the authors have access to the ages of males in their dataset, and probably of historical males too. Can any analysis be done of the effect of paternal age on CNV mutation rate? If there is a relationship, has mean paternal age changed significantly in recent times in Iceland and might this contribute to the discrepancy between the genealogical and evolutionary rates (I.231-2)?

7) In the Abstract (I.31-33) we see the CNV genealogical mutation rate given as 2.34×10^{-3} and contrasted with a 'substitution' rate of 4.42×10^{-4} . This is confusing, as most readers will think of the latter as the base substitution rate, which it is not. Some other way to express this is needed. Similar clarification is needed on pages 10-11 where the term 'substitution rate' is used.

8) The work here to examine association of Y CNV with fertility via offspring number seems reasonable, although andrologists will be unhappy at the absence of semen parameters. The other work on phenotypic traits seems parenthetical and not well justified. The list of traits is long, but it would be a great surprise if any of these were influenced by CNV of what are essentially testis-specific genes. Also, any effect is likely to be subtle, as evidenced by the Y-chromosomal influences on traits other than fertility that have been claimed to date. Furthermore, if I understand it correctly, this analysis was done only for the yellow amplicon (I.322), and not for others. This analysis needs better justification if it is to be included. The statement in I.255-8 certainly seems strong given the evidence presented here.

9) It would help some readers to have a bit more context on Y-CNVs, and pointing them to Massaia & Xue's 2017 review would be a good idea (doi:10.1007/s00439-017-1788-5). Also, the 2019 study on CNV involving P8 (doi:10.1186/s13059-019-1816-y) is relevant here and should be cited and integrated.

Minor points

The writing in the main text is good. The writing in the supplementary text is non-idiomatic in places and could do with a grammar check or read-through by a native English speaker. Also, the Table legends in supplementary information are sometimes absent or inadequate.

Throughout the manuscript, gene names and the names of genetic loci (e.g. AZFc) should be italicized; pseudoautosomal is not hyphenated.

I.48-49: Please write 'male-specific region of the Y chromosome (MSY)' as is conventional.

I.277: remove commas from dates here.

Figure 1: The text (lines 51-52) states that AZFc includes part of P3, but the bar in Figure 1 does not do this. Please explain the significance of the red and black coloured genes, and the grey shading in part A (centromere?). The Figure lacks a scale in Mb.

Part D is not self-explanatory. Is there a better way to graph these data that requires less active reasoning on the part of the reader?

Figure 1 legend: The description of part C as presented is not correct – it appears to show copy number of some of the palindromes. The part E description refers to F, but should this be D?

Table S3: Explain that these numbers are copy numbers.

Table S4: Presumably variation in copy number means duplication or deletion? Explain what the 'breakpoint' categories mean in the legend.

Table S5: In the top row, text should read '4.0-5.0'. Explain the numbers in the diagonal.

Table S10: what is 'th10'?

RESPONSE TO REVIEWERS

The original comments are in black, and the responses are in blue.

Reviewer #1 (Remarks to the Author):

This paper studies the evolution of Y-chromosome amplicons and could be a sequel to Repping et al. 2006, Nat Genet 38:463 and Teitz et al. 2018, Am J Hum Genet 103:261. The authors have access to an unparalleled and powerful dataset, including more than 11,000 male genomes sequenced to 30X coverage (15X for the Y). However, the authors neglect to indicate which sequencing platform was used (e.g., Illumina), which should be clarified. The number of samples is an order of magnitude greater than in Teitz et al., 2018, which included samples representing Y chromosomes from entire human population. This current study includes samples from one small population, Iceland, and many of the subjects are closely related, which allows the investigation of mutational processes at a much finer scale. The results confirm the conclusions of Teitz et al., 2018, as the great majority of Y chromosomes have the same amplicon copy numbers as the reference Y chromosome, presumably as the result of selection against amplicon copy number changes.

Response: We agree with the reviewer and have provided more information about how the data were generated.

Unfortunately, the authors' knowledge of the relevant human Y chromosome literature appears to be quite limited and superficial. The authors need to become better acquainted with the relevant literature and reframe the paper accordingly. I list below numerous specific examples:

We thank the reviewer for this advice. We have added references to the relevant literature suggested by the reviewer and other studies published both before and after the initial submission.

- Ln. 45: "In light of [the Y-chromosome's degeneration], it seems surprising that many of the genes on the human Y chromosome are essential for male fertility (Krausz and Casamonti 2017)." There is no existing literature on the Y chromosome, dating back decades, that indicates that the presence of male fertility genes on the Y chromosome would be unexpected in some way. In fact, quite the opposite is true.

We did not intend that to be inferred but can see how this can easily be done and have changed accordingly.

- Ln. 47: Ampliconic genes should be described as "testis-specific or testis-biased" instead of "testis-expressed." Most genes in the genome are expressed in testes (as well as in other tissues).

We agree and have changed accordingly.

- Ln. 50: “The male-specific Y (MSY) region contains eight key palindromes, with a well-established numerical nomenclature: P1 to P8 (Krausz and Casamonti 2017).” The reference for palindrome nomenclature should be Skaletsky et al. 2003, Nature 423:825.

We have included the original reference, thanks.

- There is an additional misrepresentation of the history of MSY nomenclature in T “We classified amplicon CN genotypes using a system of nomenclature devised to describe recurrent deletions and duplications in the AZFc region of the Y chromosome (reviewed in (NavarroCosta et al. 2010), Figure 1F). This nomenclature provides information about the location of deletion or duplication breakpoints, by listing the first letter of the amplicon colour label (g for green, r for red, etc. because it was originally detected with coloured probes using FISH.)” The reference for this nomenclature should be Kuroda-Kawaguchi et al. 2001, Nature Genetics 29:279. The color names assigned to amplicons do not correspond to FISH probes.

We appreciate the reviewer’s insights on this matter and have changed as suggested.

- Ln. 56: “The inverted-repeat structure of the palindromes promotes pairing between their arms, which provides a mechanism for the rescue of deleterious mutations in the ampliconic genes through gene conversion (Rozen et al. 2003; Hallast et al. 2013; Skov et al. 2017). This pseudo-diploid pairing of palindrome arms can also result in structural mutations, such as deletions and duplications, which are recurrent in human populations (Skov et al. 2017; Lucotte et al. 2018; Teitz et al. 2018; Ye et al. 2018), especially in the AZFc region.” This statement shows a lack of understanding of the mutational process under study. Deletions and duplications arise from ectopic recombination between direct repeats on the Y chromosome. Intrachromatid pairing/recombination between opposing palindrome arms, which are inverted repeats, result in typically harmless inversions. However, interchromatid recombination between opposing palindrome arms can result in deleterious isodicentric chromosome formation (Lange et al. 2009, Cell 138:855). The authors should clarify that recurrent deletions on the Y chromosome arise from recombination between direct repeats, i.e., distinct copies of the same palindrome.

We agree that concepts appear mixed up in our writing even though that was not our intention. We now explicitly state what is known about the large scale structural mutations in addition to the fine-scale changes within palindromes.

- Ln. 103: “The grey amplicon is located outside P1 and was omitted from our analysis.” The gray amplicon is actually part of P1.

We have changed the wording.

- Ln. 104: “The blue amplicon (167,703 bp) and P7 (23,149 bp) were omitted because their CN genotypes could not be called accurately, primarily due to relatively small size.” Only b1, which is an incomplete blue amplicon, is 168 kb; the other three blue amplicons are 230 kb. In Teitz et al. 2018, the number of blue amplicons was successfully defined.

We thank the reviewer for pointing out the inaccuracy in our explanation for leaving out the blue amplicon. Although Teitz et al. 2018 could accurately define the number of blue

amplicons, we found our estimates did not meet our criteria to be included confidently in our study. Indeed, as can be seen in figures S2 and S15, our CN estimates for the blue amplicon show more variance within CN clusters than other amplicons, such that there is too little separation between the expected CN clusters, therefore a lot of individuals would be assigned to an ambiguous CN. We argue that neighbor amplicons (teal and yellow) with cleaner CN estimates were better suited to capture the CN variations. Indeed, we observe that almost all of the duplication and deletion events concern whole palindrome arms, therefore adding the blue amplicon would very unlikely add information to our study. However, this was not clear from our writing and we have now changed the wording accordingly.

- This study does not include (or even mention) the one non-reproductive phenotype associated with AZFc deletions – testicular cancer (Nathanson et al. 2005, Am J Hum Genet 77:1034). This study includes a large group of samples with the *gr/gr* deletion, which has been linked to testis cancer, presenting an excellent opportunity to investigate the frequency of testis cancer in these men and potentially corroborate the existing link.

This is a very good suggestion. We have now explicitly tested for association with testicular cancer. We do not find an association. However, only 51 cases overlap with CN genotypes, so we are underpowered to detect weak association. In the case of *gr/gr* carriers we have 1 case and 501 controls, whereas for carriers of the reference CN we have 47 cases and 10467 controls, (see below). We now comment on this in the revised manuscript.

	controls	cases	tot	propCIMean	propCILow	propCIHigh
ref	10467	47	10514	4.47E-03	3.32E-03	5.99E-03
grgr_del	501	1	502	1.99E-03	1.04E-04	1.28E-02
grgr_dup	148	1	149	6.71E-03	3.50E-04	4.24E-02
grgrdelb2b4dup	21	0	21	0	0	1.92E-01
b2b3_del	80	1	81	1.23E-02	6.45E-04	7.63E-02
b2b3_dup	21	0	21	0	0	1.92E-01
b1b3	1	0	1	0	0	9.45E-01
b2b4_dup	6	0	6	0	0	4.83E-01
AZFc	2	0	2	0	0	8.02E-01
ambiguous	81	0	81	0	0	5.64E-02
unknown	146	1	147	6.80E-03	3.55E-04	4.30E-02

This paper also has serious design flaws and omissions, which are listed below:

- A critical aspect of a study measuring copy number variation is the choice of single-copy sequence for control. In Teitz et al. 2018, the control sequence was selected carefully, comprising several long stretches of Y-specific sequence from the parts of the Y unlikely to be deleted or duplicated. These intervals also served as controls for each other, so that if any one of them was deleted in some samples, it would not skew all the counts. This design could also be used in this study, but instead the authors come up with a scheme focused on the genomic sequence of single-copy Y genes as a control. The problem is that this equates to numerous short intervals, many of which can be deleted or duplicated. They also do not represent

absolutely Y-specific sequence since they have homologs on the X chromosome. The inclusion of TGF2LY and PCDH11Y from the Y chromosome's X-transposed region, which is 99% identical to the X, almost guarantees mismapping of some reads.

For normalization, we mapped the reads to an artificial Y chromosome containing as a control region the single copy genes sequences. At the same time, the reads were mapped to the X chromosome, ensuring that reads mapping preferably to X homologs would map to the X. Therefore, each bam file was only mapped once. We calculated the median coverage for each of the Y chromosome single copy genes, and then obtained an overall estimate of single copy depth as the median of those medians. Individuals with single copy depth lower than 6 reads were removed from further analyses.

The reviewer expressed concern about the length of the control region we used in comparison to Teitz et al. 2018. Put all together, the single copy genes that we used for normalization amount to more than 2Mb (2,089,261 to be exact). In Teitz et al., a 1Mb single copy region of the Y chromosome (position 14500000-15500000) was used for normalization of copy number. In parallel, four single copy regions of the Y chromosome were used as negative controls to check that they are single copies, but as far as we can see they were not used for normalization.

In order to further validate the CN estimates used in our study, we have now compared them with independently derived CN estimates for the yellow amplicon, based on a different method applied to a subset of 9162 individuals mapped independently to the entire build 38 reference genome (Gudbjartsson et al, 2015). We used lastz to obtain a list of paralogous positions in the yellow amplicon and used them to calculate, for each male, the per-position depth for y1 and y2 combined. The relative depth (rdepth) for each position in each male was obtained using as a denominator the average per-position sequence depth across 8,974,352 positions from the X-degenerate regions. Our new CN estimate was then based on the average of means or medians of rdepth in 1000bp windows across the yellow amplicon. A comparison of these CN estimates with those used in the manuscript revealed a mismatch rate of 0.003 for the median and 0.004 for the mean, where a mismatch was recorded when our independent estimate of rdepth differed by more than 0.5 from our original estimate (see Table S6). It seems very unlikely that both sets of CN estimates would be wrong in the same way, given that they are based on different read mapping, different estimates of single copy number and a different method for obtaining the y1 + y2 per-position sequence depth.

- The data in this study (as in Teitz et al. 2018) allows only the estimation of amplicon copy number; inferences about amplicon arrangements are speculative. Therefore, all conclusions should be based solely on amplicon numbers. Some of the known recurrent rearrangements do remove or duplicate whole palindromes, but most do not, and this kind of analysis is not capable of distinguishing between whole palindromes and multiple copies of single palindrome arms. Nonetheless, the authors discuss rearrangements in terms of palindrome number (e.g., ln. 198-199 and Table 2 in the main text; sections 5 and 8 in Supplementary Information).

We respectfully disagree with the reviewer's statement, we do not make inferences about amplicon rearrangements other than those that can be observed though copy number changed.

- The authors validate their method by analyzing individuals closely related to the individuals in the study, or by using independently obtained sequence from the same individuals. They get very similar results and consider this proof of the validity of the method. However, this only proves consistency. To validate their method, they would need to use an independent method to estimate copy number in samples with variations, such as FISH or quantitative PCR. If additional experiments cannot be done, the authors could process some samples from Teitz et al. 2018 through their pipeline and compare the results. Unlike the raw data from this study, the data sets from Teitz et al. 2018 are identified and publicly available.

Unfortunately, we are not able to perform FISH experiments for the samples used in the whole genome sequencing. We also note that a completely different method to evaluate CN is perhaps not necessary, given that differential sequence depth is commonly used for this purpose in next generation sequencing data. The identical results from biological replicates (of the same individual with different tissues) does prove consistency which is often the most troublesome in investigations like this. What it does not prove is that some individual-specific mapping artifact has affected the results. This is why we chose a novel approach to validate copy number by genotyping of variants in palindromic regions (for the first time, we believe). These results are very clear and they also do provide insight into the specific change that actually occurred. Also, as detailed in to previous point, we added a new validation analysis, with independently derived CN estimates for the yellow amplicon, based on a different method applied to a subset of 9162 individuals mapped independently to the entire build 38 reference genome.

- Ln. 251: The authors' discussion of the CN substitution rate is confusing. They need to clarify whether they are talking about copy number variation of amplicons or substitution rate within amplicons.

This has now been clarified.

- There is no information included about the data used in this study aside from the source: the deCODE Genetics genealogical database. Useful information to include would be sample IDs and criteria for selecting subjects (e.g., are they all of the men in their database, or only a subset?).

It is indeed all males with whole genome sequencing data available at the initiation of the analyses (2015), thus no selection. We have now made this clear. Sample IDs cannot be included for data privacy reasons.

- It is understandable that the raw data may be not available for privacy reasons, but the primary results are not even provided. The authors should include coverage estimates and copy number calls, including amplicons and control intervals, for all samples.

Icelandic law and the regulations of the Icelandic Data Protection Authority prohibit the release of individual-level and personally identifying data. We are actively participating in multiple meta-analyses based on our data and are in collaboration with groups at over 100 international universities and institutions. Therefore, collaborations using our sequencing data are based on the release of summary-level statistics, such as effect sizes and P values for meta-analysis, or collaborators traveling to our Icelandic facilities for local data access.

Therefore, it is not possible to provide what the reviewer is asking as CN calls are genotypes.

- Repping et al. 2006 includes a list of predicted AZFc rearrangements that can occur starting from the reference organization through one, two, or three deletions/duplications (via ectopic recombination between direct repeats) or inversions (via ectopic recombination between inverted repeats). In both Repping et al. 2006 and Teitz et al. 2018, each non-reference set of amplicon numbers was compared to those predicted rearrangements, and if there were matches, the one requiring the least number of steps was chosen as the likely explanation. The authors did not perform such a comparison in this study, so many of their “non-explained” cases could in fact be explained by the predictions in Repping et al. 2006.

It is true that it is possible that some of our unknown rearrangements have already been discussed in Teitz et al., but we do not feel that such a comparison would add anything to the discussions that we have which are focused on relatively simple copy number changes, thus we have not followed this specific suggestion.

- Ln. 352: “Measurement of additional fertility-related phenotypes, such as sperm counts or sperm motility might shed more light on the impact of non-reference CN genotypes and could reveal direct effects on spermatogenesis.” This statement implies that such measurements have never been done, while in fact numerous published studies have looked at these parameters in relation to AZFc deletions (e.g. Reijo et al. 1996, Lancet 347:1290; Visser et al. 2009, Hum Reprod 24:2667). This study used number of children as a proxy for fertility status/spermatogenic output, which is a poor substitute. Their results are summarized in Table S11, where the following mutations are included: gr/gr deletion, gr/gr deletion + b2/b4 duplication, b2/b3 or g1/g3 duplication, gr/gr duplication, b2/b3 deletion, b1/b3 deletion, b2/b4 duplication, and AZFc. The “AZFc” rearrangement is not mentioned in the text, so one can assume that it stands for the full AZFc deletion, which is also known as b2/b4. They found two such deletions in their population, and the mean number of children for these men is 2, which is only slightly lower than the mean number of children for men with an intact Y: 2.44. This finding confirms the fallacy of this substitute: It is well known that AZFc deletions cause severe oligozoospermia or azoospermia, and men with AZFc deletions who father children without the use of assisted reproduction are rare enough to warrant the publication of case reports (e.g. Chang et al. 1999, Hum Reprod 14:2689; Gatta et al 2002, J Med Genet 39:E27).

Our statement was meant to explain that such measurements do not exist for the males in the deCODE data. We are aware that such measurements have been used elsewhere, as we cite some of those papers. We agree that the number of recorded children is not equivalent to sperm quality measurement as this number can be affected by fertility

treatment (which we discuss line 368). Unfortunately, sperm counts were not available for the set of males with WGS data at deCODE, whereas information about the number of recorded children was available. We argue that this is a measure of realized fertility, on which individuals are selected in the end. Furthermore, regardless of the mean number of children recorded for the two men with the full AZFc deletion, it is important to bear in mind that there are only two such individuals, which precludes a reliable statistical assessment of difference with males that have the reference CN.

- There are three large spreadsheets showing absence of correlation between AZFc region rearrangements and various non-reproductive diseases. Apparently, this is considered one of the most important conclusions of the study, hence the final sentence: “Finally, it is intriguing that sequence variation on the Y chromosome, which on one hand contain the SRY gene being responsible for sex determination, and therefore the greatest normal differences in phenotypes among humans (and other mammals), otherwise has so little impact on phenotype variation among men.” However, they neglect to consider the impact of the Y chromosome’s single-copy genes, many of which are expressed across the body and likely have phenotypic consequences when deleted or mutated. There are recurring deletions on the Y – P5(P4)/P1, AZFa, and AMELY – that remove single-copy genes, which may be associated with non-reproductive phenotypes, yet it seems that the authors have not examined this possibility.

The phrasing was imprecise, we meant that the CN variants we focused on were not associated with other phenotypes. We have rephrased.

Other minor errors (likely not a complete list):

- Fig. 1A: The ampliconic gene family HSFY, located in P4, is not shown on the lower image of the chromosome.

We have corrected this.

- Supp. table S1B: The gene name DBY should be updated to DDX3Y. The location of HSFY is incorrectly indicated as P5.

Thank you for spotting this error, we have corrected it.

Reviewer #2 (Remarks to the Author):

The manuscript by Lucotte and colleagues, "Characterizing the evolution and phenotypic impact of ampliconic Y chromosome regions", investigates the genetic diversity of the human Y-chromosomal palindromes. They have used whole-genome sequence data from a total of 11,527 Icelandic men to estimate the copy number (CN) changes in these regions, making this by far the largest study to date on the topic. They report the highest number of CN changes in AZFc region as is also previously known. Due to the nature of the data, they are able to estimate and compare the CN mutation rates derived from patrilineages and Y phylogenetic tree, and due to the obtained difference suggest that selection must be favouring the reference copy number. I think it is very important to improve our understanding about these complex Y-chromosomal regions and whether or not they affect any phenotypic traits.

Overall, I find this study to be of great interest to the scientific community. The Icelandic dataset is a great resource in addition to all else also for Y-chromosomal studies due to extensive pedigree information. I have some major comments to the authors, but assuming they can adequately address these, I believe it merits publication in Nature Communications.

We thank the reviewer for these comments.

Major comments:

1. I suggest the authors to try and re-organise the Results section so that it would have a more "logical" flow and be easier for the reader to follow, perhaps also consider shortening it (e.g. by moving some details to the methods section). Even though I very much appreciate the amount of work that has gone into producing this manuscript, it is not easy to read as it stands. For example, the first section of Results, "Population variation", second paragraph on rows 100-105, starts describing the AZFc region straight away. I would suggest the authors first give a more general description of what was found in all the studied samples across all palindromes, how many rearrangements fit with the previously known deletion/CN events and how many are novel etc, and then focus in more detail on the most variable AZFc region. The same applies for other Results sections.

Also, for the description on rows 100-105 please specify that it refers only to AZFc region. Table 1 - please add the data from Teitz et al 2018 as it is currently the most thorough study of the Y-chromosomal ampliconic regions.

We seriously considered the reviewer's suggestion to re-organise the results, but given that our primary objective was to estimate the CN mutation rate using both pedigrees and a phylogenetic tree, we ultimately concluded that the original order of the sections was the most clear way to present those results. Instead, we have tried to make transitions more smooth in the revised text and to clarify terms that might lead to confusion.

2. I have some concerns about the estimation of the phylogenetic mutation rate. Perhaps the authors could explain in more detail e.g. why specific samples were chosen for the phylogenetic tree reconstruction, why 6 (or 10, see below) mutations were chosen to separate the samples, and whether the two approaches (phylogenetic vs patrilineages) produce mutation rates that can be directly compared or if some aspects should be considered while doing so, etc.

We now provide more information about how the phylogenetic tree was produced and we provide information about the SNP variants that it is based on. When reviewing the procedures used to generate this tree, we discovered a coding error that resulted in an overestimate of the number of males with CN genotypes and therefore meioses in the tree. The correct number of males with CN genotypes is 891 and the inferred number of meioses is 83,909 based on 23,118 variable SNPs from the X degenerate regions.

Also, there are confusing discrepancies between the description in the Supplementary and main text:

In the main text, lines 237-242: "To shed further light on the long-term mutation rate, we sought to estimate the phylogenetic substitution rate based on a phylogenetic tree

constructed from 1405 Icelandic Y chromosomes based on 32,914 bi-allelic sequence variants from the X-degenerate regions (8,964,929 positions). To avoid redundancy from closely related Y chromosomes and overlap with 10 de novo mutations identified in the patriline, the subset of 1405 individuals used to produce the tree was pruned such that all pairs had at least 10 mutational differences at the X-degenerate sites."

Copy from the relevant section in Supplementary, page 10: "Two separate trees were constructed, and each was pruned such that no pair of males included in the tree has fewer than 6 mutational differences. The tree includes 2221 individuals."

Thank you for pointing out this mistake. Unfortunately, this part of the supplementary material contained out of date information. We have now made sure the information provided is correct and up to date (see also response to previous point).

3. Due to the large sample size of the study, the authors are in a great position to evaluate lineage-specific differences in CN rates and I believe this would substantially strengthen the paper. According to Figure S2, a number of major lineages (I1a1, I1a2, Q1a2, R1a1, R1b1) appear to have a good representation and include quite a few samples with identified CN changes. It would be very interesting to see if there are substantial lineage-specific differences e.g. in mutation rates, types rearranged regions etc. Perhaps the large number of samples with increased copy numbers result from a few sub-lineages where such copy numbers are fixed (e.g. I1a2 a lot of samples carry CN increases in the AZFc regions)?

Following from this, a recent paper by Hallast et al (DOI: 10.7554/eLife.65420) proposes an inversion in the AZFc region specific to haplogroup R1a1-M458 which makes these Y chromosomes more prone to subsequent deletions (e.g. gr/gr and b2/b3). It would be great to see if this can be confirmed in the current dataset. The authors can identify the carriers of R1a1-M458 chromosomes from the whole-genome sequence data and investigate if indeed these show increased levels of CN changes.

Eleven males in our decode dataset carry the derived state for the R1a1-M458 variant, four of whom form father-son pairs, with the rest being outside of patriline. None of these individuals display a gr/gr or b2/b4 deletion. Given the rather small number of individuals and the fact that we can only detect relatively large changes in copy number (and not inversions), there is unfortunately little we can do to accommodate this request.

4. Curiously, the Hallast et al study reported that close to 20% of samples with gr/gr and b2/b3 (mostly in haplogroup N) deletions had undergone secondary b2/b4 duplications, whereas the current study reports only approx. 4.2% of samples with gr/gr deletion plus a secondary b2/b4 duplication (21 out of 495, Table S11). Also, even though the number of haplogroup N1c1 samples (carrying the b2/b3 deletion) does not appear very large in the current study, none of these seem to carry secondary duplications. Perhaps the authors can comment on this discrepancy.

We read the Hallast et al. paper carefully, but could not find the statement the reviewer is referring to. They report that 3/13 gr/gr males (23.1%) in their reference set carry a b2/b4 duplication.

5. Please add a table with the exact coordinates of X-degenerate regions which were used to reconstruct the Y-phylogeny. Also, please provide details about calling and filtering of these regions and about how the Y phylogeny was reconstructed.

We have now added details to the supplementary materials about how the X degenerate regions were used for this analysis (section II.1). A supplementary table was added with the coordinates used (Table S21).

Minor comments:

- row 45: "many of the gene" should be "many of the genes"

- row 71-72, sentence: "A change of CN within palindromes typically leads to a change of CN of ampliconic genes, which could have a phenotypic impact."

I find this sentence slightly misleading. Perhaps instead of "typically" it would be better to say that "can lead to a change of copy number", since palindromes P6 and P7 do not contain genes and also partial rearrangements within palindromes can happen which do not necessarily affect the genes.

You are right, thank you. We have modified the sentence.

- row 72, sentence: "Large deletions of parts of the Y chromosome are thought to cause infertility...". I would suggest changing "are thought to cause" to "are known causes of" or something similar as I do not think there is doubt about the complete AZF (a, b, c) deletions causing male infertility.

We here kept our more cautious statement, as some cases of males with AZFc deletions that could produce sperm have been reported.

- Table S7 - please add phylogenetic marker names to the haplogroup label.

Done.

- Supplementary, page 3: 3) "Construction of the artificial Y chromosome - We first constructed a reduced Y chromosome fasta sequence composed of the proximal arm sequences for palindrome 1, 3, 4, 5, 6 and 7 of the Y chromosome, in which all annotated genes were masked."

Why was P8 excluded?

P8 was not excluded, this was a mistake in the text that we have now corrected. Thank you.

Reviewer #3 (Remarks to the Author):

This study leverages the spectacular sample sizes and genomic power of Decode, and therefore presents a very impressive dataset, in which whole-genome sequence data are used to investigate copy number variation of the Y chromosome (focusing on the ampliconic regions on Yq) in many thousands of Icelandic males whose genealogical connections are known.

The questions addressed – the rates and selective/phenotypic effects of Y-chromosomal CNVs – have been much debated over many decades and are clearly in need of further illumination through population resequencing efforts, which this study offers.

The study has taken a serious approach to the topic and presents much interesting data. However, there are several aspects that need further explanation and clarification.

1) The study addresses CNVs by normalized read-depth analysis, and uses known rearrangements of the AZFc region as an interpretative framework for this complex part of the chromosome. There are some issues here:

a) The methodology employed here is blind to inversions, which have previously been demonstrated to be important because they can generate ampliconic arrangements that are permissive for particular kinds of deletions and duplications. This fact should be stressed more clearly than it is, and some discussion of how such copy-number neutral rearrangements can be detected, and what their impact is, would be welcome. Currently quite a lot has to be assumed from the data presented here about the configuration of amplicons.

This is a very valid comment which is well supported by the work of Hallast et al 2021. We now make this clear with a reference to the Hallast paper.

b) In the AZFc region, read-depth-based CNV analysis is used to assay copy number of the yellow, green, red, teal and grey amplicons, but not the blue. The text claims that the absence of the blue repeat is because of its small size (l.105), but this does not seem a reasonable explanation, because at 168 kb it is bigger than the red repeat (127 kb) or P8 (successfully typed at only 42 kb/arm). It seems more likely that failure to type the blue repeat is due to the distribution of copy numbers (Figure S2), with single unit differences between classes much commoner than for e.g. the red repeats (mostly two unit differences). It could also be due to differences in mappability of reads between different amplicon types, I guess. Whatever the reason, the absence of CNV data for blue is a nuisance. There are three of these units in the reference sequence, and they are involved in a lot of the NAHR-driven rearrangements of the region, including the classical AZFc deletion. I would urge the authors to reconsider the data for this repeat unit – the clustering in Figure S2 looks like it could be callable; if they really cannot include it, they need to do more to convince the reader that its absence does not affect their interpretation and conclusions. For example, could some of the 228 men with non-reference CN variants (l. 147-8) be interpreted if blue repeat data were available?

This was also noted by reviewer 1 and we thus repeat our answer here:

We thank the reviewer for pointing out the inaccuracy in our explanation for leaving out the blue amplicon. Although Teitz et al. 2018 could accurately define the number of blue amplicons, we found our estimates did not meet our criteria to be included confidently in our study. Indeed, as can be seen in figures S2 and S15, our CN estimates for the blue amplicon show more variance within CN clusters than other amplicons, such that there is too little separation between the expected CN clusters, therefore a lot of individuals would be assigned to an ambiguous CN. We argue that neighbor amplicons (teal and yellow) with cleaner CN

estimates were better suited to capture the CN variations. Indeed, we observe that almost all of the duplication and deletion events concern whole palindrome arms, therefore adding the blue amplicon would very unlikely add information to our study. However, this was not clear from our writing and we have now changed the wording accordingly.

c) There is a recent study (doi: 10.7554/eLife.65420) of a rearrangement in the AZFc region that is not included in the authors' set, and perhaps should be.

Our methodology does not allow us to detect inversions, which is the nature of the rearrangement in this recent study.

2) I. 42-45: The beginning of the Introduction sets out an unconvincing argument to explain the rapid evolution of mammalian Y chromosomes – ‘driven by degeneration because the Y chromosome cannot purge itself of deleterious mutations by means of recombination, with the exception of the small PARs’. Degeneration explains the loss of genes compared to the X, but is it not more likely that the lack of any pairing constraint in the MSY is responsible for toleration of rearrangements and thus the rapid evolution? In the relatively conserved PARs pairing is necessary. If the inexorable degeneration (aka Muller’s ratchet) leads to deleterious mutations these are most likely due to be subject to negative selection and lost – some of the AZF deletions are examples of this.

We agree with the reviewer’s assessment and have changed our formulation.

3) Y chromosomes are assigned to haplogroups in the study, but I could not find anywhere a description of how this was done. Presumably SNPs were called from the sequence data and correlated to haplogroup nomenclature, but which SNPs, and which nomenclature? Some information is needed, and maybe a phylogenetic tree figure. Connected to this issue is haplogroup resolution. How were decisions made on how to split haplogroups when considering CNV? Figure S2, for example shows some haplogroups highly divided but others, e.g. R1b1, not. In this same figure, haplogroup names need more explanation. Presumably e.g. I1a means I1a* (excluding I1a1, I1a2, I1a3)? Please clarify. Also, define ‘NA’.

Haplogroup assignment was performed using HaploGrouper (Jagadeesan et al, 2020) and the ISOGG 2016 tree for classification. ‘NAs’ corresponds to individuals which had not been through the haplogroup assignment pipeline at the time of the study.

4) Based on Figure S2, haplogroup I1a2 shows markedly high copy numbers compared to most other lineages. This deserves some comment.

We have now added a comment on this in the figure legend.

5) The authors used pseudohomo/heterozygosity of variants within the yellow repeat to support their CNV calls. This is a good idea, but needs a bit more explanation for the reader. Why yellow? Is this because it’s the biggest repeat unit and therefore has most variants? These variants are referred to as ‘SNPs’ (l.214 and several other places in the ms), but this term is also used for variants that are used to define lineages, and this will be confusing. Can another

term be used to avoid this? Some other authors have used 'PSV' (paralogous sequence variant), so that could be considered.

We thank the reviewer for pointing this out and now refer to these variants as PSVs. The yellow amplicon was chosen simply because it harbors most PSVs. We have included a supplementary table listing the PSVs used in the MARF analysis (Table S5).

6) It seems likely that the authors have access to the ages of males in their dataset, and probably of historical males too. Can any analysis be done of the effect of paternal age on CNV mutation rate? If there is a relationship, has mean paternal age changed significantly in recent times in Iceland and might this contribute to the discrepancy between the genealogical and evolutionary rates (l.231-2)?

To answer this very interesting question, one possibility would have been to perform an analysis of the branch length and compare the length of the branches with mutations with the length of the branches without mutations. However, we only have data for the tip of the branches, and because we expect the effects to be small, they are very probably going to average out between branches. Because we do not feel that we can contribute a very clear and clean answer to this question, we decided to not perform this analysis.

7) In the Abstract (l.31-33) we see the CNV genealogical mutation rate given as 2.34×10^{-3} and contrasted with a 'substitution' rate of 4.42×10^{-4} . This is confusing, as most readers will think of the latter as the base substitution rate, which it is not. Some other way to express this is needed. Similar clarification is needed on pages 10-11 where the term 'substitution rate' is used.

We have changed the term substitution rate to phylogenetic mutation rate in the text to avoid any confusion.

8) The work here to examine association of Y CNV with fertility via offspring number seems reasonable, although andrologists will be unhappy at the absence of semen parameters. The other work on phenotypic traits seems parenthetical and not well justified. The list of traits is long, but it would be a great surprise if any of these were influenced by CNV of what are essentially testis-specific genes. Also, any effect is likely to be subtle, as evidenced by the Y-chromosomal influences on traits other than fertility that have been claimed to date. Furthermore, if I understand it correctly, this analysis was done only for the yellow amplicon (l.322), and not for others. This analysis needs better justification if it is to be included. The statement in l.255-8 certainly seems strong given the evidence presented here.

We respectfully disagree with the reviewer's comment on non-fertility related phenotype, and argue that our dataset provides the perfect opportunity to test, to our knowledge for the first time, if CNV in ampliconic regions could affect another phenotype than fertility.

9) It would help some readers to have a bit more context on Y-CNVs, and pointing them to Massaia & Xue's 2017 review would be a good idea (doi:10.1007/s00439-017-1788-5). Also,

the 2019 study on CNV involving P8 (doi:10.1186/s13059-019-1816-y) is relevant here and should be cited and integrated.

Thank you for these suggestions, we have now included those references in the manuscript.

Minor points

The writing in the main text is good. The writing in the supplementary text is non-idiomatic in places and could do with a grammar check or read-through by a native English speaker. Also, the Table legends in supplementary information are sometimes absent or inadequate.

Throughout the manuscript, gene names and the names of genetic loci (e.g. AZFc) should be italicized; pseudoautosomal is not hyphenated.

Done

I.48-49: Please write ‘male-specific region of the Y chromosome (MSY)’ as is conventional.

Done

I.277: remove commas from dates here.

Done

Figure 1: The text (lines 51-52) states that AZFc includes part of P3, but the bar in Figure 1 does not do this. Please explain the significance of the red and black coloured genes, and the grey shading in part A (centromere?). The Figure lacks a scale in Mb.

Part D is not self-explanatory. Is there a better way to graph these data that requires less active reasoning on the part of the reader?

We have modified figure 1 following the reviewer’s remarks and clarified the legend.

Figure 1 legend: The description of part C as presented is not correct – it appears to show copy number of some of the palindromes. The part E description refers to F, but should this be D?

Thank you for pointing that out. We have corrected our mistakes.

Table S3: Explain that these numbers are copy numbers.

Done

Table S4: Presumably variation in copy number means duplication or deletion? Explain what the ‘breakpoint’ categories mean in the legend.

Done

Table S5: In the top row, text should read ‘4.0-5.0’. Explain the numbers in the diagonal.

Thank you for noticing this typo. The numbers in the diagonal are the number of observed transmissions of Y chromosomes where the copy number did not change, so from 1 to 1 copy, from 2 to 2 copies etc. We have added this to the legend of the table.

Table S10: what is ‘th10’?

Thank you for pointing that out, it was to remind the reader of the 10 mutation threshold used to prune individuals from the phylogenetic tree. As it is unclear we have removed it.

REVIEWER COMMENTS

Reviewer #2 (Remarks to the Author):

I believe the authors have substantially improved the manuscript, made it substantially more clear and transparent, and have responded to most of the reviewers comments adequately. Not all of the reviewers comments were addressed for different reasons, but overall I find it acceptable. This is an impressive dataset and the authors seem to have analysed it carefully. It is a shame that more of the underlying (raw) data cannot be made available as this would hugely benefit the scientific community, but I am aware that this is regulated by the Icelandic laws and regulations and not up to the authors.

I have one minor comment about Figure 1a – I would strongly encourage the authors to change the colour codes of the Y-chromosomal subregions to match those originally used in Skaletsky et al 2003 (i.e. for X-transposed regions etc). It should not be difficult to do and most of the chrY community is familiar with the original colour codes for the Y regions without even having to read the legend. Please feel free to ignore the striped pattern used for Yq heterochromatic region as this is difficult to reproduce and use something else.

If other reviewers agree that the comments have been address to a sufficient level, then I would suggest accepting this work for publication in Nature Communications.

Additional comments on responses to Reviewer #1:

Reviewer #1 raised some excellent points regarding data analysis and other aspects of the paper, including quite a few points regarding accuracy/validation of the copy number calls and the choice of a single-copy sequence for control. Overall the author's response looks good to me, but there is also one point they could still address better.

The choice of single-copy regions looks good to me. The authors have used all the single-copy Y genes summing to a total of ~2Mbp – these regions are spread across the Y chromosome and even if in a small set of samples some of these are affected by deletions/duplications, I would not expect those to affect the accuracy of most copy number estimates. Additionally, they have used a different approach mapping data to hg38 and using the read depth across ~8.9 Mbp of X-degenerate regions to estimate the copy numbers of the 'yellow' amplicons and obtained a very similar estimate to their original approach.

A few of the comments from Reviewer #1 are on the validation/coverage estimates/copy number calls. I understand that the genotype information cannot be made available due to Icelandic laws, but Reviewer #1 is raising an excellent point about using the 1000 Genomes Project sequence data used by Teitz et al. 2018 which is publicly available, to see if/how well the copy number estimates match between the current MS and Teitz et al. 2018. Unfortunately the authors have completely ignored this point in their response. Teitz et al. 2018 used the low coverage Illumina data, but by now high coverage Illumina data is also available for the 1000 Genomes Project samples and the authors could choose a small number of samples with different copy number estimates to see how concordant the results are. This said – it is possible that the estimates from Teitz et al. 2018 are also not 100% accurate (especially since they are based on low-coverage Illumina data), but nevertheless this would offer a good comparison especially since genotype/copy number estimates from this study are not made available. Such an analysis using a small number of samples from the 1000 Genomes Project dataset would not be a huge effort and would improve the reliability of the MS.”

Reviewer #3 (Remarks to the Author):

The authors have done a careful job of responding to reviewers’ comments and my remaining comments are only minor.

I. 45: no hyphen in pseudoautosomal (also in Figure 1 legend)

I.107: ‘extent’ is better than ‘amount’ here

I.147: ‘similarity’, not ‘identity’ (which is by definition 100%)

I.150: As Reviewer 1 points out, of the blue amplicons only b1 is 168 kb in length; this should be indicated here, or b1 specified.

I.210: ‘independent’, not ‘independently’

I.212: This line contains the first reference to the X-degenerate regions, but these are nowhere introduced or defined. Perhaps at least write ‘the single-copy X-degenerate regions’, and cite Skaletsky et al., and refer to Figure 1A where the regions are indicated.

II.215: ‘owing to their carrying’

I.216-7: There is something missing in this sentence – it does not make sense as written.

I.234: 'genotype-verified': I'm guessing this means 'autosomal genotype-verified', and perhaps this should be stated.

I.272: 'meioses' x 2

II.485-7: This sentence needs rewriting.

Generally, the English in the figure legends is less secure than in the main text and could do with polishing.

Figure 2 legend: '2 copies of the yellow amplicon'

Please write for clarity: 'The phylogenetic tree was constructed from individuals with at least 10 mutational differences at X-degenerate sites'.

Additional comments on responses to Reviewer #1 in attached document.

RESPONSE TO REVIEWERS

The original comments are in black, and the responses are in blue.

Reviewer #3 comments are in red.

Reviewer #1 (Remarks to the Author):

This paper studies the evolution of Y-chromosome amplicons and could be a sequel to Repping et al. 2006, Nat Genet 38:463 and Teitz et al. 2018, Am J Hum Genet 103:261. The authors have access to an unparalleled and powerful dataset, including more than 11,000 male genomes sequenced to 30X coverage (15X for the Y). However, the authors neglect to indicate which sequencing platform was used (e.g., Illumina), which should be clarified. The number of samples is an order of magnitude greater than in Teitz et al., 2018, which included samples representing Y chromosomes from entire human population. This current study includes samples from one small population, Iceland, and many of the subjects are closely related, which allows the investigation of mutational processes at a much finer scale. The results confirm the conclusions of Teitz et al., 2018, as the great majority of Y chromosomes have the same amplicon copy numbers as the reference Y chromosome, presumably as the result of selection against amplicon copy number changes.

Response: We agree with the reviewer and have provided more information about how the data were generated.

Reviewer #3 comment: This is OK.

Unfortunately, the authors' knowledge of the relevant human Y chromosome literature appears to be quite limited and superficial. The authors need to become better acquainted with the relevant literature and reframe the paper accordingly. I list below numerous specific examples:

We thank the reviewer for this advice. We have added references to the relevant literature suggested by the reviewer and other studies published both before and after the initial submission.

Reviewer #3 comment: OK.

- Ln. 45: "In light of [the Y-chromosome's degeneration], it seems surprising that many of the genes on the human Y chromosome are essential for male fertility (Krausz and Casamonti 2017)." There is no existing literature on the Y chromosome, dating back decades, that indicates that the presence of male fertility genes on the Y chromosome would be unexpected in some way. In fact, quite the opposite is true.

We did not intend that to be inferred but can see how this can easily be done and have changed accordingly.

Reviewer #3 comment: OK.

- Ln. 47: Ampliconic genes should be described as "testis-specific or testis-biased" instead of "testis-expressed." Most genes in the genome are expressed in testes (as well as in other tissues).

We agree and have changed accordingly.

Reviewer #3 comment: OK.

- Ln. 50: "The male-specific Y (MSY) region contains eight key palindromes, with a well-established numerical nomenclature: P1 to P8 (Krausz and Casamonti 2017)." The reference for palindrome nomenclature should be Skaletsky et al. 2003, Nature 423:825.

We have included the original reference, thanks.

Reviewer #3 comment: OK.

- There is an additional misrepresentation of the history of MSY nomenclature in T “We classified amplicon CN genotypes using a system of nomenclature devised to describe recurrent deletions and duplications in the AZFc region of the Y chromosome (reviewed in (NavarroCosta et al. 2010), Figure 1F). This nomenclature provides information about the location of deletion or duplication breakpoints, by listing the first letter of the amplicon colour label (g for green, r for red, etc. because it was originally detected with coloured probes using FISH.)” The reference for this nomenclature should be Kuroda-Kawaguchi et al. 2001, Nature Genetics 29:279. The color names assigned to amplicons do not correspond to FISH probes.

We appreciate the reviewer’s insights on this matter and have changed as suggested.

Reviewer #3 comment: I’m not sure I agree with Reviewer #1 here, but it’s no big deal.

- Ln. 56: “The inverted-repeat structure of the palindromes promotes pairing between their arms, which provides a mechanism for the rescue of deleterious mutations in the ampliconic genes through gene conversion (Rozen et al. 2003; Hallast et al. 2013; Skov et al. 2017). This pseudo-diploid pairing of palindrome arms can also result in structural mutations, such as deletions and duplications, which are recurrent in human populations (Skov et al. 2017; Lucotte et al. 2018; Teitz et al. 2018; Ye et al. 2018), especially in the AZFc region.” This statement shows a lack of understanding of the mutational process under study. Deletions and duplications arise from ectopic recombination between direct repeats on the Y chromosome. Intrachromatid pairing/recombination between opposing palindrome arms, which are inverted repeats, result in typically harmless inversions. However, interchromatid recombination between opposing palindrome arms can result in deleterious isodicentric chromosome formation (Lange et al. 2009, Cell 138:855). The authors should clarify that recurrent deletions on the Y chromosome arise from recombination between direct repeats, i.e., distinct copies of the same palindrome.

We agree that concepts appear mixed up in our writing even though that was not our intention. We now explicitly state what is known about the large scale structural mutations in addition to the fine-scale changes within palindromes.

Reviewer #3 comment: I think the authors just did not express themselves very well here, rather than misunderstanding the mutation process. However, the way the authors have rephrased this now actually makes matters worse; they say:

‘Base pairing of non-orthologous copies of ampliconic genes can lead to structural mutations, such as deletions and duplications, which are recurrent in human populations (Skov *et al.* 2017; Lucotte *et al.* 2018; Teitz *et al.* 2018; Ye *et al.* 2018), especially in the AZFc region.’

‘Base pairing’ is inappropriate, as is ‘non-orthologous’, and the whole sentence seems rather meaningless. I suggest this text for transmission to the authors.

In addressing Reviewer #1’s comments in ll.78-81 of the revised text, the authors have not improved the text. Please write instead of this sentence: ‘Ectopic pairing and recombination between direct repeats within ampliconic regions can lead to structural mutations, such as deletions and duplications, which are recurrent in human populations (refs), especially in the AZFc region.’

- Ln. 103: “The grey amplicon is located outside P1 and was omitted from our analysis.” The gray amplicon is actually part of P1.

We have changed the wording.

Reviewer #3 comment: OK.

- Ln. 104: “The blue amplicon (167,703 bp) and P7 (23,149 bp) were omitted because their CN

genotypes could not be called accurately, primarily due to relatively small size.” Only b1, which is an incomplete blue amplicon, is 168 kb; the other three blue amplicons are 230 kb. In Teitz et al. 2018, the number of blue amplicons was successfully defined.

We thank the reviewer for pointing out the inaccuracy in our explanation for leaving out the blue amplicon. Although Teitz et al. 2018 could accurately define the number of blue amplicons, we found our estimates did not meet our criteria to be included confidently in our study. Indeed, as can be seen in figures S2 and S15, our CN estimates for the blue amplicon show more variance within CN clusters than other amplicons, such that there is too little separation between the expected CN clusters, therefore a lot of individuals would be assigned to an ambiguous CN. We argue that neighbor amplicons (teal and yellow) with cleaner CN estimates were better suited to capture the CN variations. Indeed, we observe that almost all of the duplication and deletion events concern whole palindrome arms, therefore adding the blue amplicon would very unlikely add information to our study. However, this was not clear from our writing and we have now changed the wording accordingly.

Reviewer #3 comment: The authors have addressed the issue of why they did not call blue amplicon CN, but they have not addressed Reviewer #1’s first comment, that only b1, which is an incomplete blue amplicon, is 167,703 bp; the other three blue amplicons are 230 kb in size. They need to write in l.150 something like, ‘the blue amplicon (three out of four of which are ~230 kb in size)’.

- This study does not include (or even mention) the one non-reproductive phenotype associated with AZFc deletions – testicular cancer (Nathanson et al. 2005, Am J Hum Genet 77:1034). This study includes a large group of samples with the gr/gr deletion, which has been linked to testis cancer, presenting an excellent opportunity to investigate the frequency of testis cancer in these men and potentially corroborate the existing link.

This is a very good suggestion. We have now explicitly tested for association with testicular cancer. We do not find an association. However, only 51 cases overlap with CN genotypes, so we are underpowered to detect weak association. In the case of gr/gr carriers we have 1 case and 501 controls, whereas for carriers of the reference CN we have 47 cases and 10467 controls, (see below). We now comment on this in the revised manuscript.

	controls	cases	tot	propCIMean	propCILow	propCIHigh
ref	10467	47	10514	4.47E-03	3.32E-03	5.99E-03
grgr_del	501	1	502	1.99E-03	1.04E-04	1.28E-02
grgr_dup	148	1	149	6.71E-03	3.50E-04	4.24E-02
grgrdelb2b4dup	21	0	21	0	0	1.92E-01
b2b3_del	80	1	81	1.23E-02	6.45E-04	7.63E-02
b2b3_dup	21	0	21	0	0	1.92E-01
b1b3	1	0	1	0	0	9.45E-01
b2b4_dup	6	0	6	0	0	4.83E-01
AZFc	2	0	2	0	0	8.02E-01
ambiguous	81	0	81	0	0	5.64E-02
unknown	146	1	147	6.80E-03	3.55E-04	4.30E-02

This paper also has serious design flaws and omissions, which are listed below:

- A critical aspect of a study measuring copy number variation is the choice of single-copy sequence for control. In Teitz et al. 2018, the control sequence was selected carefully, comprising several long stretches of Y-specific sequence from the parts of the Y unlikely to be deleted or duplicated. These intervals also served as controls for each other, so that if any one of them was deleted in some samples, it would not skew all the counts. This design could also be used in this study, but instead the authors come up with a scheme focused on the genomic

sequence of single-copy Y genes as a control. The problem is that this equates to numerous short intervals, many of which can be deleted or duplicated. They also do not represent absolutely Y-specific sequence since they have homologs on the X chromosome. The inclusion of TGF2LY and PCDH11Y from the Y chromosome's X-transposed region, which is 99% identical to the X, almost guarantees mismapping of some reads.

For normalization, we mapped the reads to an artificial Y chromosome containing as a control region the single copy genes sequences. At the same time, the reads were mapped to the X chromosome, ensuring that reads mapping preferably to X homologs would map to the X. Therefore, each bam file was only mapped once. We calculated the median coverage for each of the Y chromosome single copy genes, and then obtained an overall estimate of single copy depth as the median of those medians. Individuals with single copy depth lower than 6 reads were removed from further analyses.

Reviewer #3 comment: The authors write here 'At the same time, the reads were mapped to the X chromosome, ensuring that reads mapping preferably to X homologs would map to the X,' which I don't understand at all – it does not seem to make sense, and mapping to the X is not mentioned either in the main text or in supplementary information. This needs clarification.

The reviewer expressed concern about the length of the control region we used in comparison to Teitz et al. 2018. Put all together, the single copy genes that we used for normalization amount to more than 2Mb (2,089,261 to be exact). In Teitz et al., a 1Mb single copy region of the Y chromosome (position 14500000-15500000) was used for normalization of copy number. In parallel, four single copy regions of the Y chromosome were used as negative controls to check that they are single copies, but as far as we can see they were not used for normalization.

Reviewer #3 comment: OK.

In order to further validate the CN estimates used in our study, we have now compared them with independently derived CN estimates for the yellow amplicon, based on a different method applied to a subset of 9162 individuals mapped independently to the entire build 38 reference genome (Gudbjartsson et al, 2015). We used lastz to obtain a list of paralogous positions in the yellow amplicon and used them to calculate, for each male, the per-position depth for y1 and y2 combined. The relative depth (rdepth) for each position in each male was obtained using as a denominator the average per-position sequence depth across 8,974,352 positions from the X-degenerate regions. Our new CN estimate was then based on the average of means or medians of rdepth in 1000bp windows across the yellow amplicon. A comparison of these CN estimates with those used in the manuscript revealed a mismatch rate of 0.003 for the median and 0.004 for the mean, where a mismatch was recorded when our independent estimate of rdepth differed by more than 0.5 from our original estimate (see Table S6). It seems very unlikely that both sets of CN estimates would be wrong in the same way, given that they are based on different read mapping, different estimates of single copy number and a different method for obtaining the y1 + y2 per-position sequence depth.

Reviewer #3 comment: This is OK. One point from Reviewer #1 that the authors have not addressed is their inclusion of the genes TGF2LY and PCDH11Y among the 'single copy' gene collection. As the reviewer points out, these are extremely similar between the X and the Y because the X to Y transposition event was very recent. Read mismapping to the X seems very likely, so in effect, these genes might appear to have a copy number of 2 in males. The authors should rebut this point or comment on it in the text.

- The data in this study (as in Teitz et al. 2018) allows only the estimation of amplicon copy number; inferences about amplicon arrangements are speculative. Therefore, all conclusions should be based solely on amplicon numbers. Some of the known recurrent rearrangements do remove or duplicate whole palindromes, but most do not, and this kind of analysis is not capable of distinguishing between whole palindromes and multiple copies of single palindrome arms. Nonetheless, the authors discuss rearrangements in terms of palindrome number (e.g., In. 198-199 and Table 2 in the main text; sections 5 and 8 in Supplementary

Information).

-

We respectfully disagree with the reviewer's statement, we do not make inferences about amplicon rearrangements other than those that can be observed though copy number changed.

Reviewer #3 comment: OK.

- The authors validate their method by analyzing individuals closely related to the individuals in the study, or by using independently obtained sequence from the same individuals. They get very similar results and consider this proof of the validity of the method. However, this only proves consistency. To validate their method, they would need to use an independent method to estimate copy number in samples with variations, such as FISH or quantitative PCR. If additional experiments cannot be done, the authors could process some samples from Teitz et al. 2018 through their pipeline and compare the results. Unlike the raw data from this study, the data sets from Teitz et al. 2018 are identified and publicly available.

Unfortunately, we are not able to perform FISH experiments for the samples used in the whole genome sequencing. We also note that a completely different method to evaluate CN is perhaps not necessary, given that differential sequence depth is commonly used for this purpose in next generation sequencing data. The identical results from biological replicates (of the same individual with different tissues) does prove consistency which is often the most troublesome in investigations like this. What it does not prove is that some individual-specific mapping artifact has affected the results. This is why we chose a novel approach to validate copy number by genotyping of variants in palindromic regions (for the first time, we believe). These results are very clear and they also do provide insight into the specific change that actually occurred. Also, as detailed in to previous point, we added a new validation analysis, with independently derived CN estimates for the yellow amplicon, based on a different method applied to a subset of 9162 individuals mapped independently to the entire build 38 reference genome.

Reviewer #3 comment: OK.

- Ln. 251: The authors' discussion of the CN substitution rate is confusing. They need to clarify whether they are talking about copy number variation of amplicons or substitution rate within amplicons.

This has now been clarified.

Reviewer #3 comment: OK.

- There is no information included about the data used in this study aside from the source: the deCODE Genetics genealogical database. Useful information to include would be sample IDs and criteria for selecting subjects (e.g., are they all of the men in their database, or only a subset?).

It is indeed all males with whole genome sequencing data available at the initiation of the analyses (2015), thus no selection. We have now made this clear. Sample IDs cannot be included for data privacy reasons.

Reviewer #3 comment: OK.

- It is understandable that the raw data may be not available for privacy reasons, but the primary results are not even provided. The authors should include coverage estimates and copy number calls, including amplicons and control intervals, for all samples.

Icelandic law and the regulations of the Icelandic Data Protection Authority prohibit the release of individual-level and personally identifying data. We are actively participating in multiple meta-analyses based on our data and are in collaboration with groups at over 100 international universities and institutions. Therefore, collaborations using our sequencing data are based on the release of summary-level statistics, such as effect sizes and P values for meta-analysis, or collaborators traveling to our Icelandic facilities for local

data access.

Therefore, it is not possible to provide what the reviewer is asking as CN calls are genotypes.

Reviewer #3 comment: OK.

- Repping et al. 2006 includes a list of predicted AZFc rearrangements that can occur starting from the reference organization through one, two, or three deletions/duplications (via ectopic recombination between direct repeats) or inversions (via ectopic recombination between inverted repeats). In both Repping et al. 2006 and Teitz et al. 2018, each non-reference set of amplicon numbers was compared to those predicted rearrangements, and if there were matches, the one requiring the least number of steps was chosen as the likely explanation. The authors did not perform such a comparison in this study, so many of their “non-explained” cases could in fact be explained by the predictions in Repping et al. 2006.

It is true that it is possible that some of our unknown rearrangements have already been discussed in Teitz et al., but we do not feel that such a comparison would add anything to the discussions that we have which are focused on relatively simple copy number changes, thus we have not followed this specific suggestion.

Reviewer #3 comment: OK.

- Ln. 352: “Measurement of additional fertility-related phenotypes, such as sperm counts or sperm motility might shed more light on the impact of non-reference CN genotypes and could reveal direct effects on spermatogenesis.” This statement implies that such measurements have never been done, while in fact numerous published studies have looked at these parameters in relation to AZFc deletions (e.g. Reijo et al. 1996, Lancet 347:1290; Visser et al. 2009, Hum Reprod 24:2667). This study used number of children as a proxy for fertility status/spermatogenic output, which is a poor substitute. Their results are summarized in Table S11, where the following mutation are included: gr/gr deletion, gr/gr deletion + b2/b4 duplication, b2/b3 or g1/g3 duplication, gr/gr duplication, b2/b3 deletion, b1/b3 deletion, b2/b4 duplication, and AZFc. The “AZFc” rearrangement is not mentioned in the text, so one can assume that it stands for the full AZFc deletion, which is also known as b2/b4. They found two such deletions in their population, and the mean number of children for these men is 2, which is only slightly lower than the mean number of children for men with an intact Y: 2.44. This finding confirms the fallacy of this substitute: It is well known that AZFc deletions cause severe oligozoospermia or azoospermia, and men with AZFc deletions who father children without the use of assisted reproduction are rare enough to warrant the publication of case reports (e.g. Chang et al. 1999, Hum Reprod 14:2689; Gatta et al 2002, J Med Genet 39:E27).

Our statement was meant to explain that such measurements do not exist for the males in the deCODE data. We are aware that such measurements have been used elsewhere, as we cite some of those papers. We agree that the number of recorded children is not equivalent to sperm quality measurement as this number can be affected by fertility treatment (which we discuss line 368). Unfortunately, sperm counts were not available for the set of males with WGS data at deCODE, whereas information about the number of recorded children was available. We argue that this is a measure of realized fertility, on which individuals are selected in the end. Furthermore, regardless of the mean number of children recorded for the two men with the full AZFc deletion, it is important to bear in mind that there are only two such individuals, which precludes a reliable statistical assessment of difference with males that have the reference CN.

Reviewer #3 comment: OK.

- There are three large spreadsheets showing absence of correlation between AZFc region rearrangements and various non-reproductive diseases. Apparently, this is considered one of the most important conclusions of the study, hence the final sentence: “Finally, it is intriguing that sequence variation on the Y chromosome, which on one hand contain the SRY gene being responsible for sex determination, and therefore the greatest normal differences in

phenotypes among humans (and other mammals), otherwise has so little impact on phenotype variation among men.” However, they neglect to consider the impact of the Y chromosome’s single-copy genes, many of which are expressed across the body and likely have phenotypic consequences when deleted or mutated. There are recurring deletions on the Y – P5(P4)/P1, AZFa, and AMELY – that remove single-copy genes, which may be associated with non-reproductive phenotypes, yet it seems that the authors have not examined this possibility.

The phrasing was imprecise, we meant that the CN variants we focused on were not associated with other phenotypes. We have rephrased.

Reviewer #3 comment: OK.

Please find below the responses (indicated in blue) to the reviewers' comments (indicated in black).

REVIEWER COMMENTS

Reviewer #2 (Remarks to the Author):

I believe the authors have substantially improved the manuscript, made it substantially more clear and transparent, and have responded to most of the reviewers comments adequately. Not all of the reviewers comments were addressed for different reasons, but overall I find it acceptable. This is an impressive dataset and the authors seem to have analysed it carefully. It is a shame that more of the underlying (raw) data cannot be made available as this would hugely benefit the scientific community, but I am aware that this is regulated by the Icelandic laws and regulations and not up to the authors.

I have one minor comment about Figure 1a – I would strongly encourage the authors to change the colour codes of the Y-chromosomal subregions to match those originally used in Skaletsky et al 2003 (i.e. for X-transposed regions etc). It should not be difficult to do and most of the chrY community is familiar with the original colour codes for the Y regions without even having to read the legend. Please feel free to ignore the striped pattern used for Yq heterochromatic region as this is difficult to reproduce and use something else.

Thank you for your suggestion, it is a very good idea that we have followed.

If other reviewers agree that the comments have been addressed to a sufficient level, then I would suggest accepting this work for publication in Nature Communications.

Additional comments on responses to Reviewer #1:

Reviewer #1 raised some excellent points regarding data analysis and other aspects of the paper, including quite a few points regarding accuracy/validation of the copy number calls and the choice of a single-copy sequence for control. Overall the author's response looks good to me, but there is also one point they could still address better.

The choice of single-copy regions looks good to me. The authors have used all the single-copy Y genes summing to a total of ~2Mbp – these regions are spread across the Y chromosome and even if in a small set of samples some of these are affected by deletions/duplications, I would not expect those to affect the accuracy of most copy number estimates. Additionally, they have used a different approach mapping data to hg38 and using the read depth across ~8.9 Mbp of X-degenerate regions to estimate the copy numbers of the 'yellow' amplicons and obtained a very similar estimate to their original approach.

A few of the comments from Reviewer #1 are on the validation/coverage estimates/copy number calls. I understand that the genotype information cannot be made available due to Icelandic laws, but Reviewer #1 is raising an excellent point about using the 1000 Genomes Project sequence data used by Teitz et al. 2018 which is publicly available, to see if/how well the copy number estimates match between the current MS and Teitz et al. 2018. Unfortunately the authors have completely ignored this point in their response. Teitz et al. 2018 used the low coverage Illumina data, but by now high coverage Illumina data is also available for the 1000 Genomes Project samples and the authors could choose a small number of samples with different copy number estimates to see how concordant the results are. This said – it is possible that the estimates from Teitz et al. 2018 are also not 100% accurate (especially since they are

based on low-coverage Illumina data), but nevertheless this would offer a good comparison especially since genotype/copy number estimates from this study are not made available. Such an analysis using a small number of samples from the 1000 Genomes Project dataset would not be a huge effort and would improve the reliability of the MS.”

We thank the reviewer for this comment. Although it required a fair amount of data download and additional processing, we accept that this is likely the most effective way to alleviate concerns about the consistency of CN calls in our study with those of the Teitz et al. study.

We now provide additional validation of our CN calling, based on our rdepth analysis was performed on 13 individuals from the 1000 Genomes Project, who were examined in the Teitz et al. (2018) study. Their copy number estimates range from 1-6 copies. The following summary table (Table 1) displays complete consistency between our estimates and those of Teitz et al. (2018). We sincerely hope that this analysis satisfies the reviewer’s concerns. We note that there are many other indicators of the reliability of our CN calls, for example the almost complete consistency of CN genotypes within patriline (with the exception of the *de novo* mutations).

Table 1- Comparison of Teitz et al. CN estimates and our CN estimate for 13 individuals from the 1000 Genomes project, representing different haplogroups and CN estimates.

ID	Haplogroup	Teitz et al. CN estimate	Our CN estimate
HG00159	I1	2	1.964
HG00182	N1	1	1.053
HG00183	N1	1	1.001
HG00242	R1b	2	1.993
HG00244	R1b	2	2.013
HG00290	N1	1	0.987
HG00338	N1	1	1.003
HG00345	I1	2	2.043
HG00371	N1	1	1.014
HG01867	O2	5	5.085
HG03445	E1b	6	6.036
NA12546	G2	4	3.918
NA20758	T	3	2.953

Reviewer #3 (Remarks to the Author):

The authors have done a careful job of responding to reviewers' comments and my remaining comments are only minor.

I. 45: no hyphen in pseudoautosomal (also in Figure 1 legend)

I.107: 'extent' is better than 'amount' here

I.147: 'similarity', not 'identity' (which is by definition 100%)

I.150: As Reviewer 1 points out, of the blue amplicons only b1 is 168 kb in length; this should be indicated here, or b1 specified.

I.210: 'independent', not 'independently'

For the five points above: thank you, we changed the text accordingly.

I.212: This line contains the first reference to the X-degenerate regions, but these are nowhere introduced or defined. Perhaps at least write 'the single-copy X-degenerate regions', and cite Skaletsky et al., and refer to Figure 1A where the regions are indicated.

Thank you for your suggestion, we changed the text.

II.215: 'owing to their carrying'

Done.

I.216-7: There is something missing in this sentence – it does not make sense as written.

Thank you, we modified it.

I.234: 'genotype-verified': I'm guessing this means 'autosomal genotype-verified', and perhaps this should be stated.

Done.

I.272: 'meioses' x 2

Done.

II.485-7: This sentence needs rewriting.

Thank you, we modified it.

Generally, the English in the figure legends is less secure than in the main text and could do with polishing.

Thank you for your comment. We have extensively re-written the figure legends, which indeed needed polishing.

Figure 2 legend: '2 copies of the yellow amplicon'

Please write for clarity: 'The phylogenetic tree was constructed from individuals with at least 10 mutational differences at X-degenerate sites'.

Done.

Additional comments on responses to Reviewer #1.

Reviewer #3 comment: I think the authors just did not express themselves very well here, rather than misunderstanding the mutation process. However, the way the authors have rephrased this now actually makes matters worse; they say:

'Base pairing of non-orthologous copies of ampliconic genes can lead to structural mutations, such as deletions and duplications, which are recurrent in human populations (Skov et al. 2017; Lucotte et al. 2018; Teitz et al. 2018; Ye et al. 2018), especially in the AZFc region.'

'Base pairing' is inappropriate, as is 'non-orthologous', and the whole sentence seems rather meaningless. I suggest this text for transmission to the authors.

In addressing Reviewer #1's comments in II.78-81 of the revised text, the authors have not improved the text. Please write instead of this sentence: 'Ectopic pairing and recombination between direct repeats within ampliconic regions can lead to structural mutations, such as

deletions and duplications, which are recurrent in human populations (refs), especially in the AZFc region.'

Thank you for your suggestions, we changed the text accordingly line 58.

Reviewer #3 comment: The authors have addressed the issue of why they did not call blue amplicon CN, but they have not addressed Reviewer #1's first comment, that only b1, which is an incomplete blue amplicon, is 167,703 bp; the other three blue amplicons are 230 kb in size. They need to write in I.150 something like, 'the blue amplicon (three out of four of which are ~230 kb in size).'

Thank you, we have modified the text accordingly.

Reviewer #3 comment: The authors write here 'At the same time, the reads were mapped to the X chromosome, ensuring that reads mapping preferably to X homologs would map to the X,' which I don't understand at all – it does not seem to make sense, and mapping to the X is not mentioned either in the main text or in supplementary information. This needs clarification.

Thank you for pointing out this missing information. We have indeed mapped against the artificial Y chromosome and the X chromosome to attract reads that would preferentially map to the X homologs. We have added this information in the supplementary material, section I.5.

Reviewer #3 comment: This is OK. One point from Reviewer #1 that the authors have not addressed is their inclusion of the genes TGF2LY and PCDH11Y among the 'single copy' gene collection. As the reviewer points out, these are extremely similar between the X and the Y because the X to Y transposition event was very recent. Read mismapping to the X seems very likely, so in effect, these genes might appear to have a copy number of 2 in males. The authors should rebut this point or comment on it in the text.

We checked this issue for one individual. It is correct that the distribution of the median read depth (coverage) of TGF2LY and PCDH11Y is significantly higher than for the rest of the single copy genes (Table 2, 79 and 90 reads compared to a median of 65 reads for the other genes). However, since we are using the median of the median read depth of all 17 single copy genes to standardize the median read depth across positions for our CN estimation of the target regions, the inclusion of these two genes has a minimal impact on our CN estimation. The current median of the median read depth for the 17 single copy genes is 66, if we exclude TGF2LY and PCDH11Y that number becomes 65. Table 3 displays the CN median and mean estimation across amplicons using both numbers. Using our method of clustering for obtaining discrete copy numbers, it has no impact on CN final estimates.

Table 2- Median coverage for each single copy genes and median of the medians with and without the genes TGIF2LY and PCDH11TY

gene	ZFY	TMSB4Y	EIF1AY	USP9Y	PRKY	AMELY	RPS4Y2	RPS4Y1	SRY	TGIF2LY	PCDH11Y	NLGN4Y	KDM5D	UTY	TBL1Y	DBY	TXLNGY	median	Median without blue genes
Median coverage	64	63	66	68	63	64	63	65	65	79	90	68	68	67	65	67	68	66	65

Table 3- CN estimation using all the single copy genes and excluding TGIF2LY and PCDH11Y (CN median 2).

amplicon	N called sites	median	sd	CN median	CN median 2	Difference
blue	120001	263	42.134	3.985	4.046	0.061
teal	17762	131	23.327	1.985	2.015	0.031
yellow	415059	134	24.504	2.030	2.062	0.031
green	263560	201	32.775	3.045	3.092	0.047
red	62583	260	47.437	3.939	4.000	0.061